

# Drivers of rapid geomagnetic variations at high latitudes

Liisa Juusola[1], Ari Viljanen[1], Andrew P. Dimmock[2], Mirjam Kellinsalmi[1], Audrey Schillings[3], and James M. Weygand[4]

[1]Finnish Meteorological Institute, Helsinki, Finland
[2]Swedish Institute of Space Physics, Uppsala, Sweden
[3]Department of Physics, Umeå University, Umeå, Sweden
[4]Department of Earth, Planetary, and Space Sciences, University of California Los Angeles, Los Angeles, CA, USA

**Correspondence:** L. Juusola (liisa.juusola@fmi.fi)

**Abstract.** We have examined the most intense external (magnetospheric and ionospheric) and internal (induced) $|d\boldsymbol{H}/dt|$ (amplitude of the 10 s time derivative of the horizontal geomagnetic field) events observed by the high-latitude International Monitor for Auroral Geomagnetic Effects (IMAGE) magnetometers between 1994 and 2018. While the most intense external $|d\boldsymbol{H}/dt|$ events at adjacent stations typically occurred simultaneously, the most intense internal (and total) $|d\boldsymbol{H}/dt|$ events were

5 more scattered in time, most likely due to the complexity of induction in the conducting ground. The most intense external $|d\boldsymbol{H}/dt|$ events occurred during geomagnetic storms, among which the Halloween storm in Oct 2003 featured prominently, and drove intense geomagnetically induced currents (GIC). Events in the prenoon local time sector were associated with sudden commencements (SC) and pulsations, and the most intense $|d\boldsymbol{H}/dt|$ values were driven by abrupt changes in the eastward electrojet due to solar wind dynamic pressure increase or decrease. Events in the premidnight and dawn local time sectors

were associated with substorm activity, and the most intense $|d\boldsymbol{H}/dt|$ values were driven by abrupt changes in the westward electrojet, such as weakening and poleward retreat (premidnight) or undulation (dawn). Despite being associated with various event types and occurring at different local time sectors, there were common features among the drivers of most intense external $|d\boldsymbol{H}/dt|$ values: pre-existing intense ionospheric currents (SC events were an exception) that were abruptly modified by sudden changes in the magnetospheric magnetic field configuration. While proper description of the fast changes during

SC events appears to require 1 s data, pulsations and substorms may be sufficiently described by 10 s $|d\boldsymbol{H}/dt|$. 1 min data, however, significantly underestimates the $|d\boldsymbol{H}/dt|$ peaks. Our results contribute towards the ultimate goal of reliable forecasts of $d\boldsymbol{H}/dt$ and GIC.

## 1 Introduction

Geomagnetically induced currents (GIC) in technological conductor networks occur commonly during space weather events

and have the potential to affect the performance of critical ground infrastructure such as electric power transmission grids. Figure 1 illustrates the chain of processes that causes geomagnetic variations and GIC. The amplitude of the time derivative of the horizontal ground magnetic field ($|d\boldsymbol{H}/dt|$) has often been used as a proxy for the geoelectric field and GIC risk (Viljanen, 1998; Viljanen et al., 2001). Geomagnetic variations observed on the ground are a sum of an "external" part due to





electric currents in the magnetosphere and ionosphere and an "internal" part due to telluric currents induced in the conducting

ground. Recently, Juusola et al. (2020) have shown that $|d\boldsymbol{H}/dt|$ is typically dominated by the internal geomagnetic variations. Because the internal part of $|d\boldsymbol{H}/dt|$ depends on the often highly structured ground conductivity (e.g., Korja et al., 2002), spatial distribution of the internal $|d\boldsymbol{H}/dt|$ is more structured than that of the external $|d\boldsymbol{H}/dt|$. An example that illustrates this is provided in Figure 2.

Fig. 2a shows the ionospheric equivalent current density ($\boldsymbol{J}$, arrows) and corresponding horizontal ground magnetic field

magnitude ($|\boldsymbol{H}|$, color), calculated by fitting the superposed magnetic field of a layer of two-dimensional Spherical Elementary Current Systems (2D SECS) (Vanhamäki and Juusola, 2020) at 90 km altitude to the measured ground geographic north ($B_x$) and east ($B_y$) components of the geomagnetic field. The down component ($B_z$) cannot be included in the fitting, because it cannot be represented in terms of ionospheric equivalent currents only. Fig. 2b–c show the ionospheric (external) equivalent current density and corresponding ground magnetic field and induced (internal) equivalent current density and corresponding

ground magnetic field, calculated by fitting the measured $B_x$, $B_y$, and $B_z$ with two layers of SECSs, one at 90 km altitude and the other at 1 m depth. The time derivatives ($dB_x(t)/dt = [B_x(t) - B_x(t-T)]/T$, where $T = 10$ s) corresponding to Fig. 2a–c are shown in Fig. 2d–f. Note that the color and arrow scales vary from panel to panel. Without the field separation (Fig. 2a, d), the measured $B_x$ and $B_y$ are perfectly reproduced by the SECS reconstruction, but $B_z$ is not. With field separation, (Fig. 2b, c, e, f) $B_x$, $B_y$, and $B_z$ are perfectly reproduced by the reconstruction. Note that this only applies to the magnetic field at

the stations. The interpolated values contain some uncertainty, and the field separation is not perfect, either (Juusola et al., 2020). The ionospheric $\boldsymbol{J}$ and corresponding ground $|\boldsymbol{H}|$ distributions, calculated without the field separation (Fig. 2a), and particularly their time derivatives (Fig. 2d), are clearly more incoherent than those calculated with the field separation (Fig. 2b, c). In order to be able to produced the highly structured $\boldsymbol{H}$ and $d\boldsymbol{H}/dt$ that in reality are caused by induced currents often located close to the surface of the earth, the ionospheric $\boldsymbol{J}$ and $d\boldsymbol{J}/dt$ calculated without the field separation (Fig. 2a, d) need

to have much stronger amplitudes than those calculated with the separation (Fig. 2b, e). Although no information about the ground conductivity has been used in the purely mathematical field separation, conductivity structures, such as the Norwegian coast line, are evident in the distribution of the induced $\boldsymbol{J}$ and $|\boldsymbol{H}|$ (Fig. 2c).

Because induction in the ground is a complicated process, external $|d\boldsymbol{H}/dt|$ may not be as good a proxy for GIC as the total $|d\boldsymbol{H}/dt|$. Namely, the geoelectric field driving GIC depends significantly on local ground conductivity, the effect of

which is also seen on the total $|d\boldsymbol{H}/dt|$. For the same reason, studying the external $|d\boldsymbol{H}/dt|$ provides more direct information on the solar wind-magnetosphere-ionosphere processes that cause intense $|d\boldsymbol{H}/dt|$ values than studying the total $|d\boldsymbol{H}/dt|$. Intense external $|d\boldsymbol{H}/dt|$ is a requirement for strong induction and GIC, although the most intense induced $|d\boldsymbol{H}/dt|$ may not occur at the same time as the most intense external $|d\boldsymbol{H}/dt|$. This is because the induced current density depends on the time history of the external $d\boldsymbol{H}/dt$ as well as the structure of the ground conductivity, which may favor certain type of driving over

others. Understanding the details of the solar wind-magnetosphere-ionosphere coupling that create intense external $|d\boldsymbol{H}/dt|$ is a requirement for developing our capability to forecast them. The very short persistence of the time derivative of the horizontal ground magnetic field ($d\boldsymbol{H}/dt$) compared to horizontal ground magnetic field ($\boldsymbol{H}$) is a challenge for forecasting $d\boldsymbol{H}/dt$ (Kellinsalmi et al., 2022). Once time series of the external $\boldsymbol{H}$ with correct $d\boldsymbol{H}/dt$ behavior can be predicted, fast modeling of



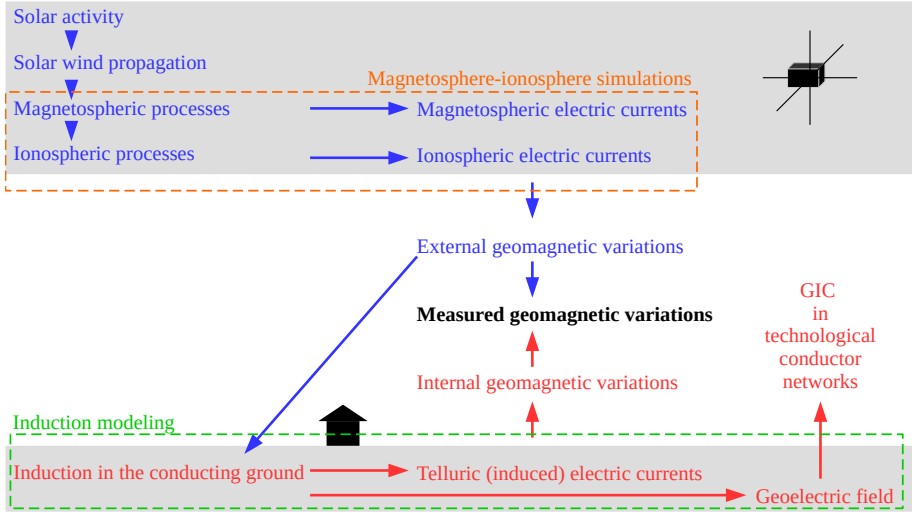

**Figure 1.** Principle of geomagnetic variations and geomagnetically induced currents (GIC): Disturbances created by solar activity propagate to the Earth and interact with the Earth's magnetosphere-ionosphere system, creating electric currents in the near-Earth space. The spatially and temporally highly varying "external" magnetic field of these currents drives induction in the conducting ground. The "internal" magnetic field of the induced telluric electric currents is superposed to the external magnetic field, producing geomagnetic variations that can be measured by magnetometers on ground or at low-orbit satellites. The relative contribution of internal and external magnetic fields depends on the distance of the measurement point to the various current systems, such that the internal contribution is the strongest on the ground and weakens with increasing altitude. The induced geoelectric field drives GIC in technological conductor systems. Global magnetosphere-ionosphere simulations can typically describe the external part of geomagnetic variations only, while induction modeling is needed for describing the electromagnetic interaction between the conducting Earth and external geomagnetic variations.

three-dimensional (3D) induction (e.g., Marshalko et al., 2021; Kruglyakov et al., 2022) and GIC (Lehtinen and Pirjola, 1985; Viljanen et al., 2012, 2014; Kelly et al., 2017; Pirjola et al., 2022) is feasible.

Intense $|d\boldsymbol{H}/dt|$ and GIC are typically attributed to substorms, sudden commencements (SCs), and geomagnetic pulsations (Viljanen et al., 1999; Rogers et al., 2020; Clilverd et al., 2021) associated with geomagnetic storms driven by interplanetary coronal mass ejections (ICMEs) and their sheaths (Huttunen et al., 2008; Kataoka and Pulkkinen, 2008). Energy input from the solar wind into the magnetosphere can be characterized by various parameters, such as the solar wind electric field or southward component of the interplanetary magnetic field (IMF), and has been found to correlate with GIC amplitudes and nightside magnetic perturbation activity (Huttunen et al., 2008; Engebretson et al., 2021). Recently, Hajra (2022) has shown that intense GIC do not occur as individual peaks but as clusters with duration of ∼5–38 h associated with intense substorm clusters, characterized by peak SuperMAG auroral electrojet index >2000 nT.

The SC signature is a sudden increase of the horizontal geomagnetic field. It occurs when an abrupt increase in the solar wind dynamic pressure, at an interplanetary shock, for example, compresses the magnetosphere, leading to an intensification of the electric currents in the magnetosphere and ionosphere (Oliveira and Samsonov, 2018).





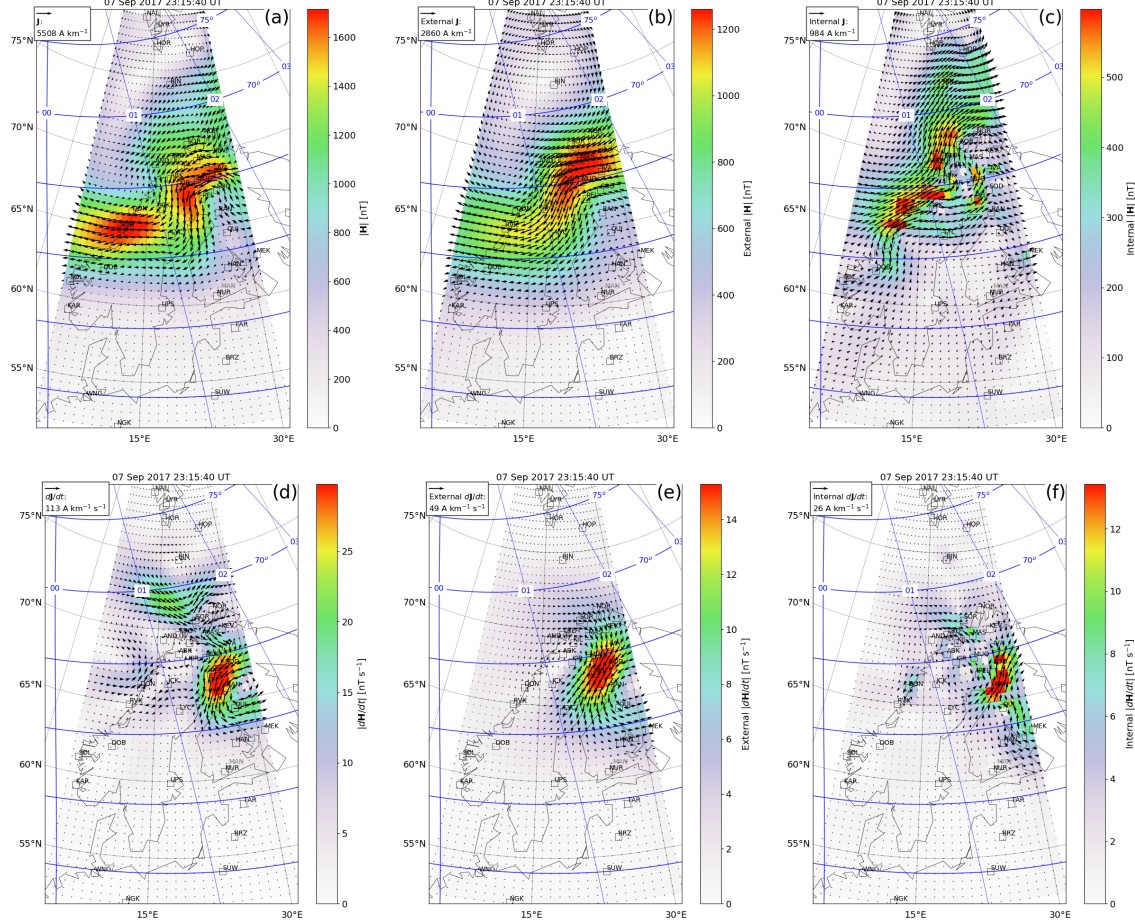

**Figure 2.** Effect of ground magnetic field separation: Ionospheric equivalent current density (arrows) and corresponding horizontal ground magnetic field, calculated by fitting the measured ground north ($B_x$) and east ($B_y$) components of the geomagnetic field with a layer of 2D Spherical Elementary Current Systems (SECS) (Vanhamäki and Juusola, 2020) at 90 km altitude (a). (b)–(c): Ionospheric (external) equivalent current density and corresponding ground magnetic field and induced (internal) equivalent current density and corresponding ground magnetic field, calculated by fitting the measured $B_x$, $B_y$, and $B_z$ with two layers of SECSs, one at 90 km altitude and the other at 1 m depth. (d)–(f): The same as (a)–(c) except for the time derivatives. Note that each panel has a different color and arrow length scale.

Ultra-low frequency waves (ULF), which are detected on the ground as geomagnetic pulsations, are a known source of intense $|d\boldsymbol{H}/dt|$. ULF waves in the magnetosphere can be caused either externally by solar wind perturbations, or internally. The most important sources of ULF waves directly driven by the solar wind are thought to be the Kelvin-Helmholtz instability and solar wind dynamic pressure pulses. Because the Kelvin-Helmholtz instability requires a shear flow, it mainly occurs at the dawn and dusk flanks of the magnetopause, especially when the solar wind speed is high (Engebretson et al., 1998).





Whereas SCs and some pulsation types can be considered to be directly driven by the solar wind, substorms are a delayed response to energy input from the solar wind into the magnetosphere. Intense $|d\boldsymbol{H}/dt|$ (exceeding 1 nT/s) occur typically during events governed by westward ionospheric currents (Viljanen et al., 2001), often during substorm onsets when the
80 amplitude of the westward electrojet (WEJ) increases rapidly (Viljanen et al., 2006b). Juusola et al. (2015a) have shown that the temporal development of $|d\boldsymbol{H}/dt|$ activity during substorms follows that of substorm currents and aurora, typically spreading from a premidnight onset region poleward, equatorward and toward the morning sector along the auroral oval and WEJ. According to Viljanen et al. (2001) and Juusola et al. (2020), the directional distribution of intense external $d\boldsymbol{H}/dt$ varies as a function of magnetic local time (MLT), developing from the predominantly north-south orientation in the premidnight
sector to east-west orientation in the 02–05 MLT sector, and back to north-south direction in the pre-noon sector. This change is strongest at stations at the latitudes of the nominal auroral oval and becomes weaker towards north and south, where $d\boldsymbol{H}/dt$ prefers north-south orientation even in the morning sector. According to Schillings et al. (2022), rapid $dB/dt$ spikes initially occur in the premidnight sector and later spread towards the morning sector. Such a sequence is correlated with the AE index and can repeat several times throughout a geomagnetic storm.

Intense $|d\boldsymbol{H}/dt|$ events are concentrated in the pre-midnight and dawn MLT sectors (Juusola et al., 2015b; Schillings et al., 2022). The pre-midnight events are most likely associated with substorms onset. Substorm onsets leading to clear bulge-type substorms typically occur around 23 hours MLT, but they are generally not observed after 03 hours MLT (e.g., Frey et al., 2004). Auroral streamers, driven by fast magnetospheric flows, can also occur in the morning sector auroral oval (e.g., Forsyth et al., 2020). In the ground magnetic field, auroral streamers typically correspond to southward propagating enhancements
of northwestward or westward ionospheric equivalent current (Juusola et al., 2009). Streamers appear to play a role in the formation of auroral omega bands (Weygand et al., 2015; Forsyth et al., 2020; Weygand et al., 2022). Omega bands tend to be associated with times of greater than average geomagnetic activity and their occurrence peaks in the 02–04 hours MLT sector, mainly during substorm recovery phases (Partamies et al., 2017). The main source of magnetic disturbances on the ground during omega bands is an undulating WEJ (Opgenoorth et al., 1983), which causes $d\boldsymbol{H}/dt$ in the east-west direction, although
the ambient field is generally southward. Geomagnetic pulsations at periods 5–40 min, called the Ps6 category (Jacobs et al., 1964), due to the undulating WEJ are associated with omega band activity (Opgenoorth et al., 1983). Omega bands have been shown to cause intense GIC (Apatenkov et al., 2020).

Pulkkinen et al. (2003a) and Dimmock et al. (2019) have suggested that GIC are primarily driven by small-scale spatio-temporal structures superimposed on the large-scale WEJ. Recent studies have demonstrated that nighttime magnetic pertur-
105 bations at high latitudes can occur in association with a range of ionospheric current systems, geomagnetic conditions, and auroral structures, and can cover large, moving regions with diameters of hundreds of km (Ngwira et al., 2018; Engebretson et al., 2019a, b; Dimmock et al., 2020; Weygand et al., 2021).

In this study, we will examine in detail five events with the most intense external $|d\boldsymbol{H}/dt|$ (10 s time resolution) observed by the International Monitor for Auroral Geomagnetic Effects (IMAGE) magnetometer network between 1994 and 2018. The
110 open question we try to approach is the details and drivers in the various processes (SCs, substorms, pulsations) that produce





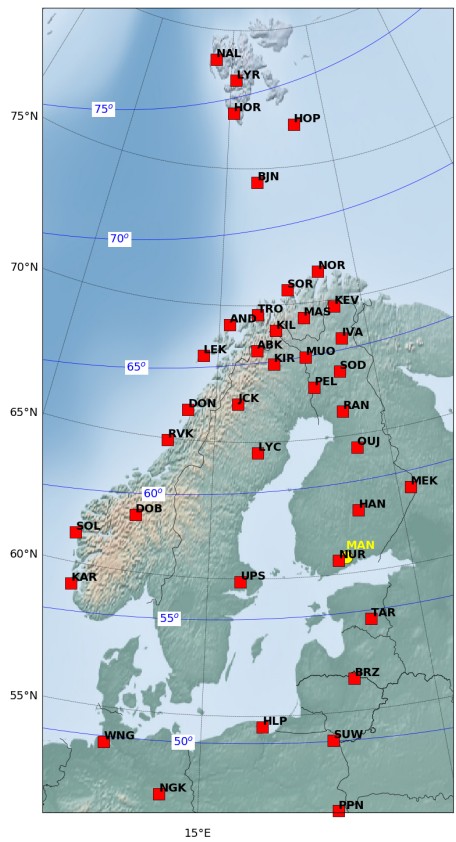

**Figure 3.** Stations of the International Monitor for Auroral Geomagnetic Effects (IMAGE) magnetometer network (red squares) in 2018 used in the current study and the natural gas pipeline close to Mäntsälä (MAN, yellow circle) where GIC are recorded. The blue curves indicate geomagnetic latitudes.

the most intense $|d\boldsymbol{H}/dt|$. The structure of the study is as follows: the data and methods are presented in Section 2, the results are presented in Section 3 and discussed in Section 4. The conlusions are summarized in Section 5.

## 2 Data and method

### 2.1 Data

We have used 10 s ground magnetic field measurements from IMAGE (https://space.fmi.fi/image/) magnetometers between 1994 and 2018. Currently, IMAGE consists of 41 stations that cover magnetic latitudes from the subauroral 47° N to the polar 75° N in an approximately two hour magnetic local time (MLT) sector. The stations included in our analysis are shown in Figure 3 and availability of data from each magnetometer is shown in Figure 4.





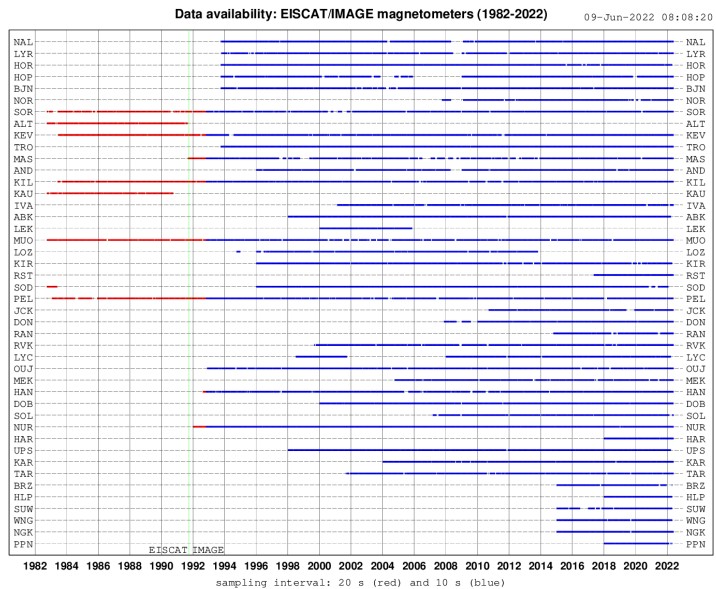

**Figure 4.** Availability of data from IMAGE magnetometers.

IMAGE data are provided in geographic coordinates and we carry out the analysis using the same coordinate system. We use
the notations $B_x$, $B_y$, and $B_z$ for the north, east and down components of the ground magnetic field. The horizontal magnetic
field vector is denoted by $\boldsymbol{H} = B_x \hat{\boldsymbol{e}}_x + B_y \hat{\boldsymbol{e}}_y$ and its amplitude by $|\boldsymbol{H}| = \sqrt{B_x^2 + B_y^2}$. Similarly, the time derivative vector and
its amplitude are $d\boldsymbol{H}/dt = dB_x/dt\hat{\boldsymbol{e}}_x + dB_y/dt\hat{\boldsymbol{e}}_y$ and $|d\boldsymbol{H}/dt| = \sqrt{(dB_x/dt)^2 + (dB_y/dt)^2}$, respectively. The measured
magnetic field is a sum of the internal and external contributions, e.g., $B_x = B_{x,internal} + B_{x,external}$. The time derivative is
calculated as $dB_x(t)/dt = [B_x(t) - B_x(t-T)]/T$, where $T = 10$ s is the time step of the data. Although geographic coordinates
are used to present the data, we have occasionally marked magnetic coordinates in the plots (e.g., the blue curves in Fig. 3).
We have used the quasi-dipole (QD) coordinates (Richmond, 1995; Emmert et al., 2010) as given by the software available at
https://apexpy.readthedocs.io/en/latest/. The code uses the 12th generation International Geomagnetic Reference Field (IGRF-
12, Thébault et al., 2015).

To assess the GIC-effectiveness of the intense external $|d\boldsymbol{H}/dt|$ events, we have used 10 s GIC recordings (https://space.
fmi.fi/gic/) in a Finnish natural gas pipeline close to the Mäntsälä (MAN) compressor station in southern Finland (60.6 N, 25.2
E) (Pulkkinen et al., 2001; Viljanen et al., 2006a). As shown in Fig. 3, MAN is located about 40 km eastward from the IMAGE
magnetometer station Nurmijärvi (NUR). 1 min SYM-H index (Iyemori and Rao, 1996), auroral electrojet indices (AE, AL,
AU, Davis and Sugiura, 1966) and solar wind data propagated to Earth's bow shock nose were extracted from NASA/GSFC's
OMNI data set through OMNIWeb (Papitashvili and King, 2020). A list of sudden commencements (SC) (Curto et al., 2007)





has been obtained from http://isgi.unistra.fr/. An SC is a sharp increase in the magnetic field north component and the SC lists
are made on the basis of visual inspection of magnetograms from five low latitude observatories.

## 2.2  Field separation and equivalent currents

We have used the two-dimensional Spherical Elementary Current System (2D SECS) method (Amm, 1997; Amm and Viljanen,
1999; Pulkkinen et al., 2003a, b; Juusola et al., 2016; Vanhamäki and Juusola, 2020; Juusola et al., 2020) to separate the
magnetic field measured at each station into internal and external parts and to derive the external (at 90 km altitude) and
internal (at 1 m depth) equivalent current densities for each epoch. Before application of the 2D SECS method, a baseline
needed to be subtracted from the data. Because most IMAGE stations are variometers, we could not use a model of the Earth's
main field. Instead, we have used the method by van de Kamp (2013) to remove the long-term baseline (including instrument
drifts, etc.), any jumps in the data, and the diurnal variation. The diurnal quiet-time magnetic field variation in the IMAGE
region is at most a few tens of nT (Sillanpää et al., 2004). We concentrate on studying large time derivatives of the horizontal
magnetic field for which this effect is insignificant.

The total (internal + external) magnetic field from the 2D SECS reconstruction is identical to the measured field at all
IMAGE stations used in the SECS reconstruction. However, separation and interpolation of the geomagnetic field between the
stations are not perfect and are affected by the density of the magnetometers as well as boundary conditions, as discussed by
Juusola et al. (2020). Nonetheless, the internal part of the separated field has been shown to follow well the known structure
of the ground conductivity (Juusola et al., 2020, see also Fig. 2c, where the coastline is prominent) and correlation between
the electrojet currents derived simultaneously from IMAGE and low-orbit satellite have been shown to significantly improve
when the separation is carried out (Juusola et al., 2016). These results indicate that, although not perfect, the separation should
be fairly reliable and worth carrying out.

A change in the station configuration can, under certain conditions, result in an artificial time derivative peak in the recon-
structed magnetic field at the nearby stations. Because of this, we have discarded any station with data gaps during a day when
processing the entire time series between 1994 and 2018 for the statistics. The time derivative has been calculated without com-
bining values from different days. This is a fairly strict approach and wastes some usable data around midnight, but ensures
that there will not be any artificial time derivative peaks due to changes in station configuration.

## 160  3  Results

Table 1 shows the universal time (UT) of the most intense total, external, and internal $|d\boldsymbol{H}/dt|$ at each IMAGE station between
1994 and 2018. MLT and the SYM-H index are also provided. The stations are listed according to decreasing geographic
latitude (see Fig. 3 for their locations). Three events with several instances in the table have been indicated with colors: the
Halloween geomagnetic storm on 29–31 Oct 2003 (blue, Panasyuk et al., 2004; Pulkkinen et al., 2005), the geomagnetic storm
on 7–8 Sep 2017 (orange, Dimmock et al., 2019), and the geomagnetic storm on 24 Nov 2001 (green, Tsurutani et al., 2015;
Kleimenova et al., 2015). Colored dots in the second column indicate station availability during the days of these events (black:





not available). For example: station NAL does not have data for the first day of Event 1 (29 Oct 2003) but has data for the last two days of Event 1 (30–31 Oct 2003), for both days of Event 2 (7–8 Sep 2017), and for Event 3 (24 Nov 2001). Table 1 updates the list of the most intense $|dH/dt|$ in 1994–2003 calculated from 10 s IMAGE data by Pulkkinen et al. (2005).

The maximum external $|dH/dt|$ values can be attributed to a handful of events. The Halloween storm in Oct 2003 is clearly the main source of the most intense $|dH/dt|$ values at the IMAGE stations, with the Sep 2017 storm generally as the replacement for those stations that did not have data from the Halloween storm. In addition to the Oct 2003 and Sep 2017 geomagnetic storms that mainly affected the stations in the middle of the IMAGE latitude range, the storm in Nov 2001 and an event in Jan 2005 caused the most intense values in northern Fennoscandia and Svalbard, respectively. The station BJN

is located on an island between Fennoscandia and Svalbard, with some distance to Fennoscandia stations in the south and Svalbard stations in the north, and had its own most intense event. The southern IMAGE station TAR, which has provided data since 2001, observed the most intense values during the "St. Patrick's day" geomagnetic storm on 17 Mar 2015 (Wu et al., 2016). The stations south of TAR are fairly new to IMAGE and only have a few years of data. The maximum external $|dH/dt|$ at IMAGE, 21.2 nT/s, was observed at KIR during a geomagnetic storm in Dec 2006. Generally, the external $|dH/dt|$ effects

of each event were limited in latitude and concentrated further south with decreasing SYM-H index values, as expected. Events associated with SCs, indicated in Table 1 by the superscript [a], were naturally an exception, with positive values of concurrent SYM-H.

The most intense internal (61.3 nT/s) and also total $|dH/dt|$ (77.0 nT/s), observed by RVK, is attributed to the SC at the beginning of the Halloween storm. However, generally the events producing the most intense internal and, consequently total,

$|dH/dt|$ values at IMAGE were more scattered in time than those that produced the maximum external values. Even during the same event, the maximum values at different stations did not typically occur at the same time. Most likely this can be attributed to the complex ground conductivity structure as well as the optimal temporal development of the external $dH/dt$ that drives induction. The intensity of the maximum external $|dH/dt|$ is fairly uniform across IMAGE, between 0.5 nT/s and 21.2 nT/s (median: 13.2 nT/s, mean: 11.8 nT/s, standard deviation: 6.3 nT/s), with some dependence on data availability (Fig. 4). The

variability in the internal (range: 0.8–61.3 nT/s, median: 22.6 nT/s, mean: 22.0 nT/s, standard deviation: 12.9 nT/s) and, consequently, total (range: 1.2–77.0 nT/s, median: 32.2 nT/s, mean: 29.8 nT/s, standard deviation: 16.3 nT/s) maximum $|dH/dt|$, on the other hand, is much larger, due to the complexity of induction in the conducting ground.

Next, we will study more closely five events that caused the most intense external $|dH/dt|$ values: 29 Oct 2003 at 06:11:50 UT (SC at the beginning of the Halloween storm), 30 Oct 2003 at 20:08:40 UT (substorm event of the Halloween storm), 24

Nov 2001 at 07:32:20 UT (storm-time pulsation event), 15 Dec 2006 at 02:07:10 UT (storm-time substorm event), and 17 Apr 1999 at 02:07:10 UT (storm-time substorm event). The last event is not represented in Table 1 but is responsible for very intense external $|dH/dt|$ values at several IMAGE stations.

### 3.1  Prenoon SC event on 29 Oct 2003 at 06:11:50 UT

On 29 Oct 2003 at 06:10:00 UT, a SC occurred, signifying the start of the Halloween storm. Figure 5e shows the SYM-H

index around this time. The SC is marked by a blue solid vertical line. At 06:11:40–06:11:50 UT (black dashed vertical line in



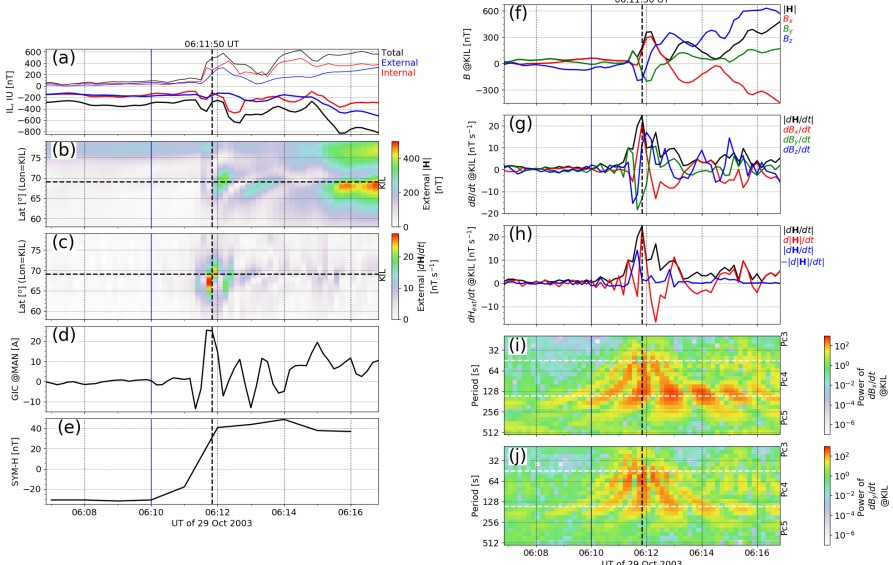

**Figure 5.** (a): Auroral electrojet indices derived from IMAGE data (Kauristie et al., 1996) $\pm 5$ min around the event on 29 Oct 2003 at 06:11:50 UT. The index derived from total (external + internal), external, and internal geographic $B_x$ from all available IMAGE stations is drawn with black, blue, and red color, respectively. The thicker curves show the lower envelope curve IL index and the thinner curves the upper envelope curve IU index. (b): Latitude profiles of external ground $|\boldsymbol{H}|$ as a function of UT along the longitude of station KIL (indicated in Fig. 6 by black vertical lines). (c): Latitude profiles of external 10 s $|d\boldsymbol{H}/dt|$ as a function of UT along the longitude of station KIL. (d): GIC in the natural gas pipe at MAN (location close to the IMAGE station NUR indicated in Fig. 6). (e): SYM-H index. (f): Geographic external $B_x$, $B_y$, and $B_z$ at station KIL. (g): $dB_x/dt$, $dB_y/dt$, $dB_z/dt$ at KIL. (h): Time derivative of the external horizontal magnetic field vector ($|d\boldsymbol{H}/dt|$), time derivative of the amplitude of the horizontal ground magnetic field vector ($d|\boldsymbol{H}|/dt$), and the difference $|d\boldsymbol{H}/dt| - |d|\boldsymbol{H}|/dt|$, which represents the contribution of the change in vector direction to $|d\boldsymbol{H}/dt|$ at KIL. (i–j): Wavelet transform of $dB_x/dt$ and $dB_y/dt$ at KIL. The period ranges of the ultra-low frequency (ULF) pulsation classes Pc3 (10–45 s), Pc4 (45–150 s) and Pc5 (150–600 s) (Jacobs et al., 1964) are shown with the white, horizontal, dashed lines. The blue vertical line indicates a SC at 06:10:00 UT and the vertical dashed black line indicates the time of the extreme external $|d\boldsymbol{H}/dt|$ observed at KIL.

Fig. 5e), several IMAGE stations measured their most intense external $|d\boldsymbol{H}/dt|$ between 1994 and 2018 (Table 1). At the same time, an intense (e.g., Hajra, 2022) GIC peak of 25 A (Fig. 5d) was recorded at MAN.

Fig. 5i–j show wavelet transforms (e.g., Torrence and Compo, 1998; Fligge et al., 1999) of external $dB_x/dt$ and $dB_y/dt$ at KIL. We use continuous wavelet transform with Morlet wavelets as given by the software available at https://pywavelets.

readthedocs.io/en/latest/ (Gregory et al., 2019). In addition to the time interval shown in Fig. 5f–g, we have included an equally long period of data before and after the interval of interest, i.e., analyzed an interval three times as long as that shown in Fig. 5f–g. The period ranges of the ultra-low frequency (ULF) pulsation classes Pc3 (10–45 s), Pc4 (45–150 s) and Pc5 (150–600 s) (Jacobs et al., 1964) are indicated with white horizontal dashed lines. The wavelet transform reveals decaying oscillations around 128 s period as well as faster variations around 06:11:50 UT.



Figure 6a shows a map of the external equivalent current density ($\boldsymbol{J}$, arrows) and the corresponding horizontal ground magnetic field magnitude ($|\boldsymbol{H}|$, color) on 29 Oct 2003 at 06:11:40 UT when the most intense external $|d\boldsymbol{H}/dt|$ between 1994 and 2018 was measured at IVA, PEL, RVK, and DOB. The time derivatives of $\boldsymbol{J}$ and $\boldsymbol{H}$ are displayed in Fig. 6b. Fig. 6c–d show the same for 06:11:50 UT, when the most intense external $|d\boldsymbol{H}/dt|$ was measured at KEV, MAS, KIL, ABK, and MUO. Fig. 6b and d illustrate the geographical extent of the $|d\boldsymbol{H}/dt|$ signature. Its temporal development explains the different times of maximum $|d\boldsymbol{H}/dt|$ at different stations. Station SOD is displayed in these plots as one of the stations strongly affected by the signature, but in Table 1 it shows missing data for 29 Oct 2003. The reason is that time derivatives for the statistics were processed one day at a time, and any station with data gaps during that day was omitted, because changes in station configuration produce artificial temporal changes in the resulting data at nearby stations. The time development of the ionospheric equivalent currents and the magnetic field they produce on the ground is further illustrated by the animation IMAGE_20031029T060650_10sec_20031029T061650.mp4, which consists of frames similar to Fig. 5a–e, 6a, and 6b on 29 Oct 2003 between 06:06:50 UT and 06:16:50 UT (10 min) with a 10 s time step.

The SC caused the appearance, intensification, and northward propagation of the eastward electrojet (EEJ), tilted slightly from geomagnetic eastward to north-eastward. The limited spatial extent indicates that this current is indeed an enhancement of the ionospheric EEJ and not an equivalent current representation of the sudden enhancement of the magnetopause currents (Fiori et al., 2014). The temporal evolution of the EEJ enhancement is illustrated in Fig. 5b and 5c, which present latitude profiles of $|\boldsymbol{H}|$ and $|d\boldsymbol{H}/dt|$ as a function of UT along the longitude of station KIL, indicated in Fig. 6 by black vertical lines. The auroral electrojet indices IL (thick curves) and IU (thin curves) derived from total (external + internal), external, and internal IMAGE data in Fig. 5a show only a modest increase in the external IU during the SC.

Time series of the geographic external magnetic field components and their time derivatives at KIL are displayed in Fig. 5f–g. In agreement with Fig. 6 and Fig. 5a–c, they show a short-lived intensification of northeastward current passing over the station. The most intense external $|d\boldsymbol{H}/dt|$ at 06:11:50 UT is associated with the appearance of the EEJ. Fig. 5h shows the change of the external horizontal magnetic field vector ($|d\boldsymbol{H}/dt|$, black), its amplitude ($d|\boldsymbol{H}|/dt$, red), and their difference ($|d\boldsymbol{H}/dt| - d|\boldsymbol{H}|/dt$, blue), representing the contribution of the change in vector direction to $|d\boldsymbol{H}/dt|$. $d|\boldsymbol{H}|/dt$ is zero if the intensity of $\boldsymbol{H}$ remains constant but its direction can still change. According to Fig. 5h, the maximum external $|d\boldsymbol{H}/dt|$ (black curve) is caused by the intensification of the EEJ (red curve), with practically no contribution from the change of direction in the current (blue curve).

Figure 7 shows the total $|d\boldsymbol{H}/dt|$ derived from 1 s, 10 s, and 1 min data at selected IMAGE stations (panels arranged according to decreasing geographic latitude) in 29–31 Oct 2003. This period includes the SC signature at the beginning, intense substorm activity at nighttime, and pulsations in the morning of 31 Oct 2003 (Panasyuk et al., 2004). The 1 s data yields the largest values, but apart from the SC signature, 10 s values are almost as large. On the other hand, amplitudes derived from 1 min data are significantly weaker. The difference between low and high time resolution data is especially pronounced at the subauroral stations OUJ, HAN, and NUR, whereas at the auroral stations KEV, MAS, KIL, IVA, MUO, and PEL and the southernmost station TAR the difference is smaller. At NUR, for example, 1 s $|d\boldsymbol{H}/dt|$ had a peak value of 179.4 nT/s on 29 Oct 2003 at 06:11:29 UT, 10 s data had a peak values of 17.7 nT/s at 06:11:50 UT, and 1 min data had a peak of 1.8 nT/s





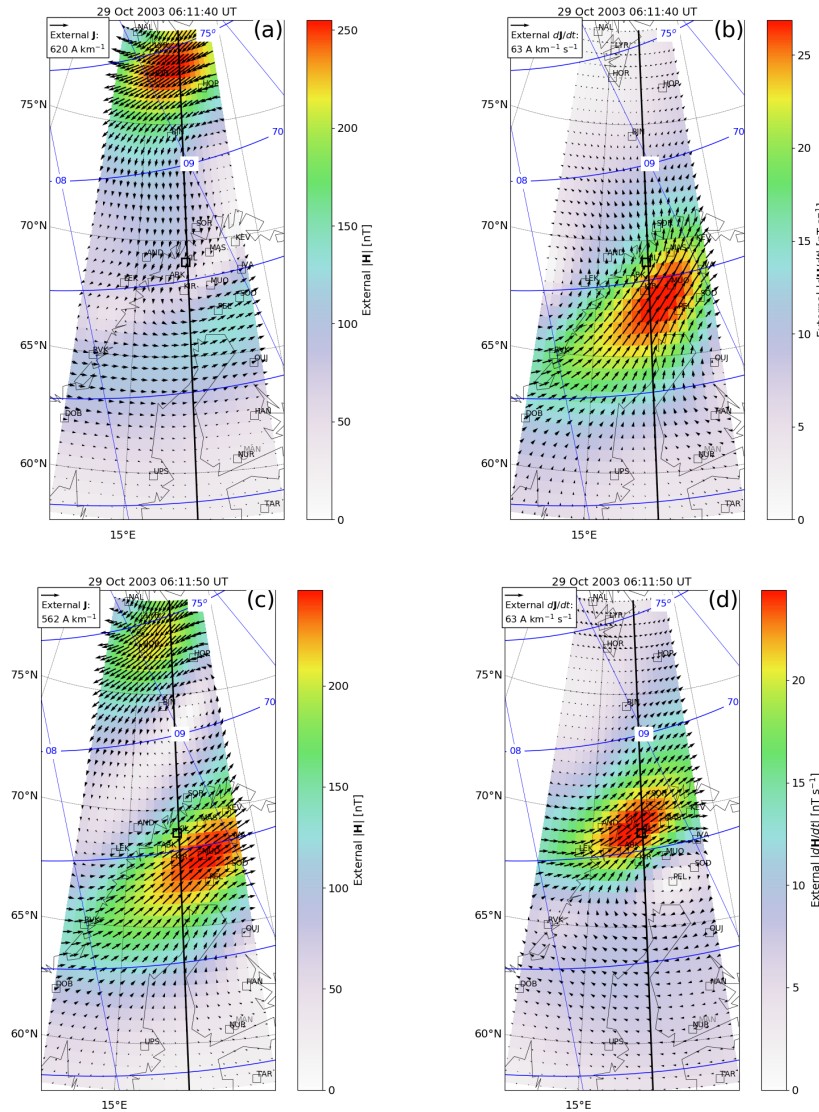

**Figure 6.** (a): External equivalent current density ($\boldsymbol{J}$, arrows) and external horizontal magnetic field on ground ($|\boldsymbol{H}|$, color) on 29 Oct 2003 at 06:11:40 UT when the largest external $|d\boldsymbol{H}/dt|$ between 1994 and 2018 was measured at stations IVA, PEL, RVK, and DOB. (b): 10 s time derivative of external $\boldsymbol{J}$ ($d\boldsymbol{J}/dt$, arrows) and of external $\boldsymbol{H}$ ($|d\boldsymbol{H}/dt|$, color). (c)–(d): The same as (a)–(b), but at 06:10:50 UT, when the the largest external $|d\boldsymbol{H}/dt|$ was measured at stations KEV, MAS, KIL, ABK, and MUO. In each panel, a black vertical line passing through the station KIL indicates the locations from which latitudes profiles are extracted to create a time series representation of the parameters in Fig. 5.

at 06:13:00 UT (Fig. 8a). NUR is located close to MAN, which showed a peak in 10 s GIC values at 06:11:40–06:11:50 UT





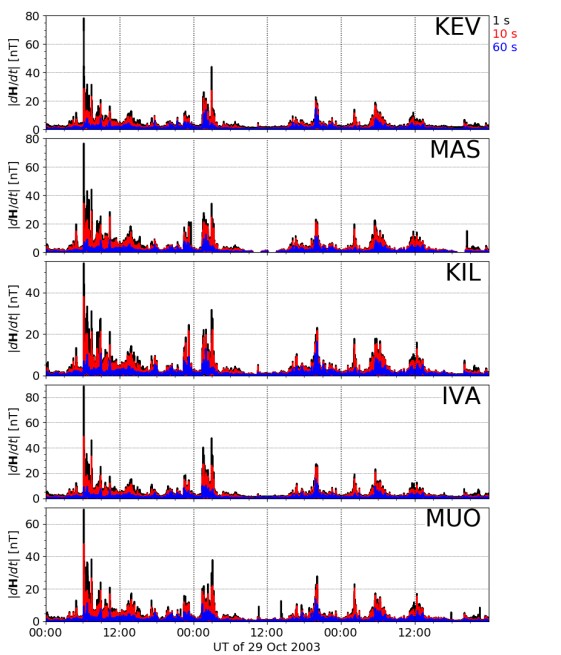
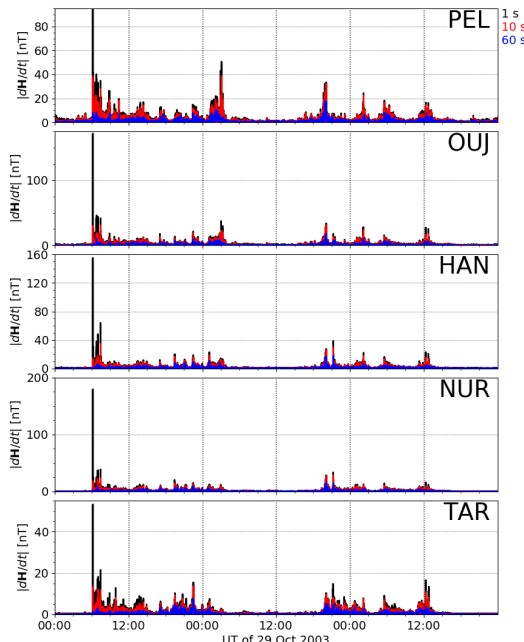

**Figure 7.** Time derivative of the total horizontal magnetic field ($|d\boldsymbol{H}/dt|$) derived from 1 s, 10 s, and 1 min data at selected IMAGE stations in 29–31 Oct 2003.

(Fig. 5d). Inspection of the total $d\boldsymbol{H}/dt$ at NUR around 06:11:50 UT (not shown) indicates that the oscillating 10 s GICs are well described by the 10 s $dB_x/dt$.

The big difference between 1 s, 10 s and 1 min values in Fig. 7 is partly compensated when considering the geoelectric field. As Pulkkinen et al. (2006) concluded, the sampling rate of the magnetic field can be lowered down to about one minute without

crucially degrading the accuracy of the modelled electric field. However, at least SCs are an obvious exception as shown in Fig. 8. Figure 8b shows the modelled geoelectric field at NUR. We used a simple two-layer ground conductivity model whose upper layer is 30 km thick and has a resistivity of 5000 Ωm, and the lower infinitely deep layer has a resistivity of 20 Ωm. For the present purpose, we can assume a plane wave field and calculate the electric field by using the 1D surface impedance. The importance of using 1 s data is clearly visible: the maximum value of the electric field related to the huge peak of $|d\boldsymbol{H}/dt|$

(Fig. 8) is very much reduced if modelled from 10 s or 1 min magnetic field. On the other hand, after 06:12 UT, there is not any more significant difference between 1 s and 10 s electric field values. For the Malmö blackout substorm on 30 Oct 2003, 10 s and even 1 min magnetic field data is fully sufficient for estimating the maximum electric field (figure not shown).



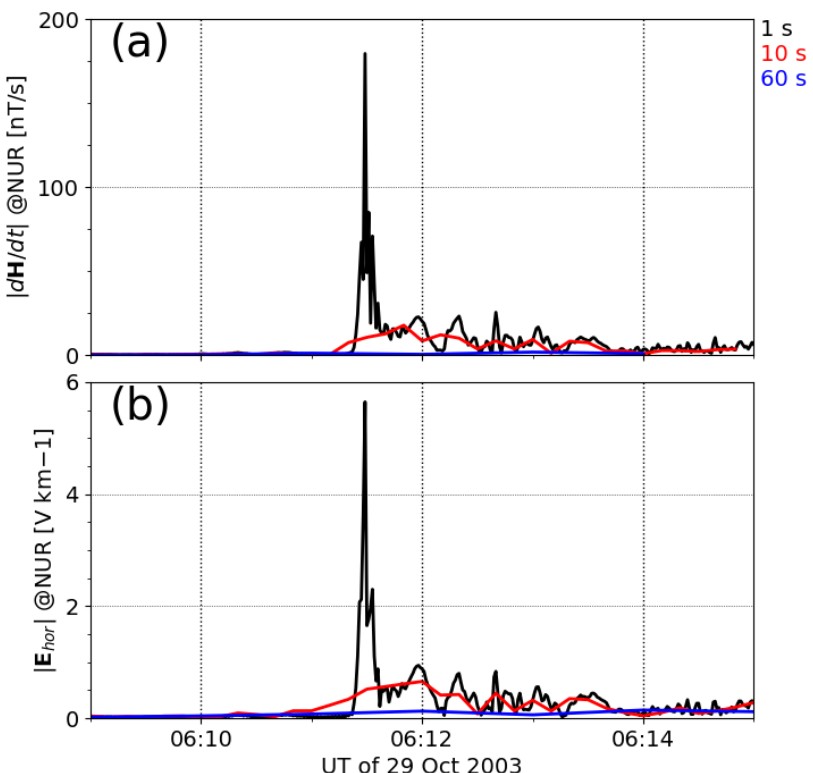

**Figure 8.** Zoom-in on the total $|d\mathbf{H}/dt|$ measured at NUR in Fig. 7, based on 1 s (black), 10 s (red), and 1 min (blue) geomagnetic data (a). Modelled geoelectric field at NUR (b).

The most intense external $|d\mathbf{H}/dt|$ event observed at LYC (Table 1) on 10 Nov 2000 at 06:28:40 UT (∼09 MLT) was also associated with a SC. Examination of the event (not shown) reveals that it shared many similar features with the SC on 29 Oct

2003. In both cases the most intense $|d\mathbf{H}/dt|$ was caused by an abrupt intensification and poleward propagation of the EEJ. Finally, we note that the appearance of a SC in the list of the largest $|d\mathbf{H}/dt|$ events is a little fortuitous. The SC on 29 Oct 2003 just happened to occur at a time when the IMAGE network was in an optimal local time sector to measure very large variations.

## 3.2 Premidnight substorm event on 30 Oct 2003 at 20:08:40 UT

Figure 9 shows a substorm event that took place in the middle of the Halloween storm, on the evening of 30 Oct 2003. The format is similar to the corresponding panels in Fig. 5 and Fig. 6c–d, but shows a time interval of 35 min instead of 10 min. MAN





(Fig. 9d) shows an intense GIC peak of $-49$ A at 20:08:20–20:08:30 UT (the sign of the GIC indicates the direction of the current in the pipe, positive eastward). According to Fig. 9, an intensification and poleward and equatorward expansion of a WEJ occurred around 19:47:00 UT, typical for a local substorm onset. At 20:08:40 UT, the latitudinally wide, strong WEJ abruptly

weakened (Fig. 9b), causing an intense time derivative signature over a long latitude range (Fig. 9c). The most intense external $|d\boldsymbol{H}/dt|$ was observed at this time at the IMAGE stations OUJ, HAN, NUR, and UPS (Table 1). These stations are located in the southern part of IMAGE (Fig. 9k–l) and were not as much affected by the (10 s) SC signature on 29 Oct 2003 as the stations in the northern part of Fennoscandia (Fig. 6). The magnetic field time series at station OUJ (Fig. 9f–h) confirm the sudden weakening of the WEJ at 20:08:40 UT, but also reveal wave activity (Fig. 9g) associated with the weakening. The wavelet transforms

of external $dB_x/dt$ and $dB_y/dt$ (Fig. 9i–j) show wave activity across a wide range of periods, but particularly around 128 s period. The time development is also illustrated by the animation IMAGE_20031030T194340_10sec_20031030T201840.mp4, which is composed of frames similar to Fig. 9a–e, 9k, and 9l on 15 Dec 2006 between 01:52:10 UT and 02:22:10 UT (30 min) with a 10 s time step.

Comparison of Fig. 9b and 9e reveals that the two intensifications of the WEJ ($\sim$19:53:00–19:58:00 UT and $\sim$20:02:00–

20:08:40 UT) coincided with enhancements of the SYM-H index of a few tens of nT. In the absence of direct solar wind drivers, increases of 20–40 nT in SYM-H after substorm onsets have been suggested to be caused by dipolarization, where the inner magnetosphere on the nightside is highly compressed, resulting in an increase in the ground $|\boldsymbol{H}|$ and SYM-H (Huang et al., 2004). The southward expansion of the WEJ that coincides with the SYM-H peaks supports this interpretation. Furthermore, the abrupt weakening and northward retreat of the WEJ at 20:08:40 UT coincides with the return of the SYM-H index back to

the level preceding the enhancement, indicating a sudden recovery from dipolarization, possible due to cessation of fast flows from a reconnection site farther downtail. It should be noted that this substorm event was very strong and produced the most intense external horizontal ground magnetic field ($|\boldsymbol{H}|$) between 1994 and 2018 at several IMAGE stations (Table 2).

Pulkkinen et al. (2005) have previously studied GIC and their relation to problems in the Swedish high-voltage power transmission system during the Halloween storm. They identified two periods during which problems in the power transmission

system were observed: 06:10–07:05 UT on 29 Oct 2003 and 19:35–20:30 UT on 30 Oct 2003. These periods include our events (SC event at 06:11:40–50 UT on 29 Oct 2003 and substorm event at 20:08:40 UT on 30 Oct 2003). Pulkkinen et al. (2005) attributed the problems in operating the Swedish system during the storm broadly to substorms, storm sudden commencement, and enhanced ionospheric convection, all of which created large and complex geoelectric fields capable of driving large GIC. Our results further specify that the largest external $|d\boldsymbol{H}/dt|$ values were caused by an abrupt intensification of an EEJ due to

the compression of the Earth's magnetosphere, and an abrupt weakening of a substorm WEJ, possibly in association with an expansion of the inner magnetosphere and the transition region between dipolar and tail-like field lines, respectively.

### 3.3    Prenoon pulsation event on 24 Nov 2001 at 07:32:20 UT

On 24 Nov 2001, a geomagnetic storm driven by an interplanetary coronal mass ejection (ICME) and its sheath occurred (Tsurutani et al., 2015). According to Tsurutani et al. (2015), two extremely intense substorms with the peak SuperMAG

auroral electrojet index of $-3839$ nT and $-3312$ nT took place at $\sim$07:00–07:50 UT and at $\sim$13:45–14:18 UT, respectively.





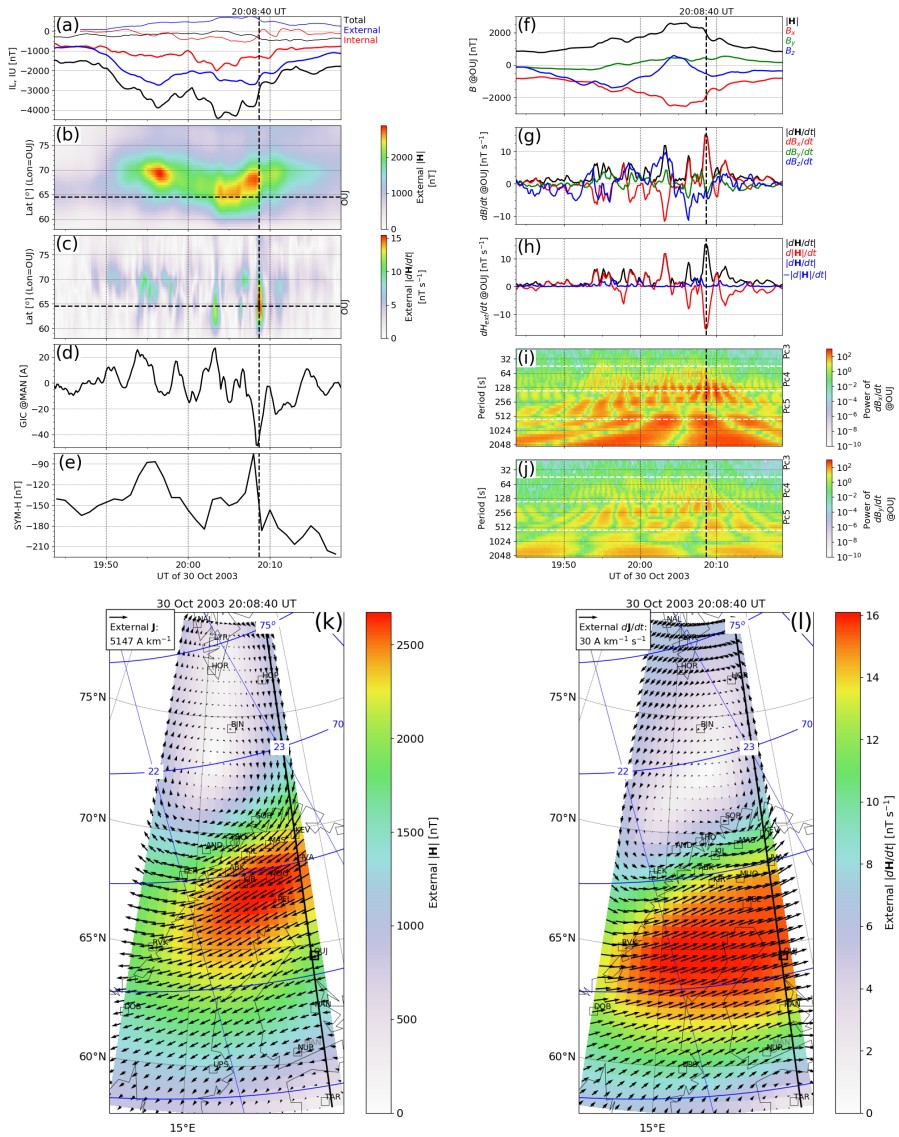

**Figure 9.** Data in a format similar to combined Fig. 5 and Fig. 6c–d from −25 min to +10 min around the event on 30 Oct 2003 at 20:08:40 UT.

Our event with intense external $|d\boldsymbol{H}/dt|$ at IMAGE occurred during the first substorm, at 07:32:20 UT. At this time, IMAGE was on the dayside, around 10 MLT. The first substorm began during a period of strong southward IMF in the sheath, but by the time of our event, IMF had turned northward. This is illustrated in Figure 10, which shows solar wind, SYM-H, and auroral electrojet index data. Timing of the effects of solar wind data on the ground always contains some uncertainty (e.g.,

uncertainty due to the propagation from the observation point to the Earth's bow shock and the delay from the bow shock to





the ionosphere), but it appears that the event may have coincided with a drop in the solar wind dynamic pressure (Fig. 10d) after a very high peak.

Figure 11 illustrates the event at IMAGE in a format similar to Fig. 9. At 07:32:20 UT, the most intense $|d\boldsymbol{H}/dt|$ was observed at SOR, TRO, AND, and LEK (Table 1). According to Fig. 11l, the intense $|d\boldsymbol{H}/dt|$ signature was concentrated
in northern Fennoscandia. Station KIL is listed as not having data on 24 Nov 2001 due to some data gaps during the day but appears in Fig. 11k–l. Fig. 11 shows that the intense $|d\boldsymbol{H}/dt|$ was associated with the EEJ that was disrupted due to wave activity. A wavelet transform of $dB_x/dt$ and $dB_y/dt$ in Fig. 11i–j indicates wave activity around 128 s period around and after 07:32:20 UT as well as faster variations in the Pc4 period range around 07:32:20 UT. The time development is also illustrated by the animation IMAGE_20011124T071720_10sec_20011124T074720.mp4, which is composed of frames similar
to Fig. 11a–e, 11k, and 11l on 24 Nov 2001 between 07:17:20 UT and 07:47:20 UT (30 min) with a 10 s time step.

Analogous to the SC event on 29 Oct 2003, it appears that the compression of the dayside magnetopause by intense dynamic pressure created an EEJ signature in the prenoon auroral ionosphere. Whereas intense $|d\boldsymbol{H}/dt|$ on 29 Oct 2003 was associated with the abrupt appearance of the EEJ, in this event intense $|d\boldsymbol{H}/dt|$ is associated with the disappearance of the EEJ as the dynamic pressure weakened. The EEJ signature did not disappear abruptly, but faded away during the time it took for the
dynamic pressure to weaken. The disappearance of the EEJ, however, was not smooth but overlaid with decaying wave activity (Parkhomov et al., 1998). This pulsation activity caused the most intense $|d\boldsymbol{H}/dt|$ by temporarily disrupting the EEJ. As can be seen in Fig. 11b–c, the pulsation signatures first appeared at lower latitudes and from there propagated northward, producing stronger variations where the ionospheric currents were stronger. The subauroral counterpart of these signatures appears to be responsible for the GIC peak of 26 A at 07:32:50 UT at MAN, close to the NUR magnetometer located at 60.5° latitude.

Another pulsation event (not shown) with the most intense (although relatively weak) external $|d\boldsymbol{H}/dt|$ in Table 1 was the event at southern IMAGE stations HLP and WNG on 20 Aug 2018 at 10:22:50–10:23:00 UT (∼12 MLT). The solar wind dynamic pressure during this event was nominal, but the solar wind speed was high, around 600 km/s.

### 3.4 Morning sector substorm event on 15 Dec 2006 at 02:07:10 UT

A SC occurred on 14 Dec 2006 at 14:14:18 UT, and was followed later by a geomagnetic storm driven by fast solar wind
and strong southward IMF (Figure 12). The event of intense external $|d\boldsymbol{H}/dt|$ at IMAGE took place at 02:07:10 UT, near the SYM-H minimum of the storm (Fig. 12e). Fig. 13 shows the event at IMAGE in a format similar to Fig. 9. The most intense external $|d\boldsymbol{H}/dt|$ observed by any IMAGE station between 1994 and 2018, 21.2 nT/s, was detected at station KIR at 02:07:10 UT (∼04 MLT). Fig. 13l shows that this was a relatively localized event peaking near KIR. Several nearby stations list this event as second largest after the Halloween storm. The event did not produce significant GIC peaks at MAN (Fig. 13d), most
likely due to the relatively localized nature of the $|d\boldsymbol{H}/dt|$ signature.

Fig. 13b shows that the most intense $|d\boldsymbol{H}/dt|$ occurred during a sequence of events during which the WEJ repeatedly intensified, jumped poleward and slowly drifted equatorward. The most intense $|d\boldsymbol{H}/dt|$ was associated with the sudden weakening (Fig. 13h) and poleward retreat of the WEJ at the end of the first of these cycles. The most intense $|d\boldsymbol{H}/dt|$ during the Halloween substorm event (Section 3.2) also occurred when the WEJ suddenly weakened and retreated poleward. The





large $dB_y/dt$ contribution to $|d\boldsymbol{H}/dt|$ (Fig. 13g, l) in this event, while the ambient WEJ remained mainly east-west oriented (Fig. 13f, k), indicates electrojet undulations, possibly associated with auroral streamer or omega band activity. Fig. 14 shows the wavelet transform of external $dB_x/dt$ and $dB_y/dt$ at KIR for the longer time interval of Fig. 12 compared to that shown in Fig. 13i–j. The Ps6 period range 5–40 min (Jacobs et al., 1964) is indicated with the white, horizontal, dashed lines. According to Fig. 14, Ps6 activity occured for several hours around 02:07:10 UT, during the time the SYM-H index slowly started to

recover after the plummet of the main phase of the storm (Fig. 12). The time development is also illustrated by the animation IMAGE_20061215T015210_10sec_20061215T022210.mp4, which is composed of frames similar to Fig. 13a–e, 13k, and 13l on 15 Dec 2006 between 01:52:10 UT and 02:22:10 UT (30 min) with a 10 s time step.

### 3.5    Morning sector substorm event on 17 Apr 1999 at 03:10:50 UT

Solar wind, SYM-H, and auroral electrojet data for our final event on 17 Apr 1999 are displayed in Fig. 15. Similar to the oth-

ers, this event took place during a geomagnetic storm that started with a SC on 16 Apr 1999 at 11:24:54 UT and was followed by a clear sheath and ICME. The solar wind speed for this event was not high, but there was a strong sustained southward IMF associated with the front part of the ICME which drove the geomagnetic storm with a minimum SYM-H of around $-100$ nT. The event at IMAGE is illustrated in Fig. 16. It took place at 03:10:50 UT (05–06 MLT), close to the time of the SYM-H minimum of the geomagnetic storm. According to Fig. 16, the intense external $|d\boldsymbol{H}/dt|$ was associated with a change of current

direction from southwestward to westward in an intensified westward electrojet. Similar to the previous event, there is a large $dB_y/dt$ contribution to $|d\boldsymbol{H}/dt|$, up-and-down motion of the WEJ, and Ps6 activity (Figure 17), which indicate WEJ undulation. The time development is also illustrated by the animation IMAGE_19990417T025550_10sec_19990417T032550.mp4, which is composed of frames similar to Fig. 16a–e, 16k, and 16l on 17 Apr 1999 from 02:55:50 UT to 03:25:50 UT (30 min) with a 10 s time step.

Other events of the most intense external $|d\boldsymbol{H}/dt|$ in Table 1 associated with an intensified, undulating WEJ were the events at DON on 08 Sep 2015 at 01:15:00 ($\sim$03 MLT), at SOD and RAN on 07 Sep 2017 at 23:15:40 UT ($\sim$02 MLT), and at JCK on 08 Sep 2017 at 00:33:20 UT ($\sim$03 MLT).

## 4    Discussion

We have examined in detail five events that were responsible for the most intense external $|d\boldsymbol{H}/dt|$ in the IMAGE region

between 1994 and 2018. All except the 17 Apr 1999 event are listed by Schillings et al. (2022) among the 27 strongest storms in 1980–2020 in terms of the number of observed intense $dB/dt$ spikes worldwide. Our events and our interpretations are summarized in Table 3.

    The five events represent all event types generally associated with intense $|d\boldsymbol{H}/dt|$ or GIC: SCs, pulsations, and substorms. All were associated with geomagnetic storms preceded by SCs. Despite the wide range of event types and occurrence MLTs,

the events share some common features: three out of the five events appeared to be directly associated with changes in the magnestospheric magnetic field configuration either due to compression (SC) or expansion (weakening of solar wind dynamic





pressure or nightside dipolarization). In the other two events, the equatorward and poleward motion of the WEJ indicates some changes in the magnetospheric magnetic field configuration as well, although the cause is not clear. Fast magnetospheric flows associated with auroral streamer or omega band activity could be possible drivers.

What is noteworthy in our five of events is that there are no substorm onsets or sudden intensifications of the WEJ among them. Viljanen et al. (2006b) have shown that substorm onsets are among the most significant drivers of intense $|d\boldsymbol{H}/dt|$. Examination of the rest of the most intense external $|d\boldsymbol{H}/dt|$ in Table 1 reveals that the most intense events at the northern IMAGE stations NOR on 08 Oct 2015 at 18:04:50 UT (∼21 MLT) and BJN on 04 Oct 1995 at 15:01:30 UT (∼18 MLT) were such intensifications. Furthermore, the events at the southern stations TAR and BRZ on 17 Mar 2015 at 17:33:50–17:35:30 UT

(∼20 MLT) and at SOL, KAR, WNG, and NGK on 08 Sep 2017 at 17:57:50–17:56:00 UT (∼19 MLT) appear to be associated with the southward expansion of subtorm currents. There can be several reasons why the more intense events do not include substorm onsets: One reason could be that induced currents play a very significant role in substorm onsets (e.g., Tanskanen et al., 2001), which would make them prominent in the examination of the total $|d\boldsymbol{H}/dt|$ but not when only the external component is analysed. Another reason could be that although substorm onsets generally produce large $|d\boldsymbol{H}/dt|$, the most

intense values require more extreme conditions: pre-existing intense ionospheric currents (the SC event was an exception) that are abruptly modified by rapid changes in the magnetospheric magnetic field configuration. Wave activity around the event can significantly enhance the $|d\boldsymbol{H}/dt|$ signature. A third possibility is that, despite the long, continuous time series (1994–2018) of analysed $|d\boldsymbol{H}/dt|$, IMAGE has simply happened to be in the wrong MLT sector during the limited number of geomagnetic storms that have produced the most intense $|d\boldsymbol{H}/dt|$ peaks worldwide. The mechanisms that produce the $|d\boldsymbol{H}/dt|$ peaks during

these storms would vary according to MLT, but globally the most intense peaks would be associated with substorm onsets in the appropriate MLT sector where IMAGE just did not happen to be located at the crucial moments.

### 4.1 Strong vs. extreme space weather events

Although the most intense events observed by IMAGE between 1994 and 2018 are certainly strong, they do not appear to be in any way exceptional compared to other strong events observed by IMAGE. This can be seen in Figure 18, which shows

a histogram of all total, internal, and external $|d\boldsymbol{H}/dt|$ values at KIR between 1994 and 2018. The most intense external 10 s $|d\boldsymbol{H}/dt|$ observed by IMAGE between 1994 and 2018, 21.2 nT/s, was measured at KIR in 15 Dec 2006 (Section 3.4). Functions in the form $a\exp^{-b|d\boldsymbol{H}/dt|}$ (e.g., Myllys et al., 2014), where $a$ and $b$ are constants, have been fitted into the data points in Fig. 18, and can describe the occurrence of the most intense values quite well. Thus, our study may not provide information on truly extreme space weather conditions comparable to the Carrington geomagnetic storm (Clauer and Siscoe,

2006; Blake et al., 2021). Such storms are not covered by our observations and may be so much stronger than the observed events that data-based extrapolations cannot describe them. On the other hand, extreme value analysis by Wintoft et al. (2016) indicates that stations north of the magnetic latitude 60° already include $|d\boldsymbol{H}/dt|$ values that are close to the expected maxima, while more equatorward stations will measure larger values in the future.



## 4.2 Forecasting $|d\boldsymbol{H}/dt|$ and GIC

As discussed by Pulkkinen et al. (2017), the horizontal geoelectric field expressed in time and space provides the key input to power grid operators to determine GIC and its impacts. A general risk analysis can be based on benchmark events, especially the most intense ones. Then it is possible to use long time-series of geomagnetic recordings and provide the modelled geoelectric field for a large set of different events (e.g., Viljanen et al., 2014). For this purpose, it is also meaningful to use geomagnetic data from other locations as long as they are from approximately the same magnetic latitudes as the region of interest. Ideally, the

external part of the variation field should be used, since it does not depend on effects due to local telluric currents (cf. Tables 1 and 2).

  Concerning forecasting ground magnetic field variations from real-time L1 solar wind observations (e.g., Balan et al., 2017), our study helps in identifying events which are likely to cause large $|d\boldsymbol{H}/dt|$ values (and large GIC): a strong SC or modification of pre-existing intense ionospheric currents by sudden changes in magnetospheric magnetic field configuration. This may

be useful when tailoring empirical forecasts based on neural networks, for example. If the existence of intense "background" ionospheric currents can be predicted, or directly seen from real-time observations, then a general alarm of higher probability of a big $|d\boldsymbol{H}/dt|$ event can be given. What obviously remains difficult is to anticipate whether there will be a trigger leading to intense $|d\boldsymbol{H}/dt|$, and where and when this will take place. If the Halloween SC had begun a few minutes later, the largest $|d\boldsymbol{H}/dt|$ values would obviously have occurred over the ocean. This shows that even if a SC can be quite definitely anticipated

from L1 observations, there is still much uncertainty in predicting precisely the most severely affected area. Challenges appear also with substorms such as that of 30 Oct 2003 causing the Malmö blackout. Similar problems are faced by first-principle simulations, which are still far from optimal performance (e.g., Kwagala et al., 2020). Recently, Dimmock et al. (2021) have shown that by increasing the spatial resolution of global magnetohydrodynamic (MHD) models, the ability to model GIC can be improved. However, substorms still remain a problem, because the simulations cannot capture the dynamics of the ionospheric

currents that drive the complex external $d\boldsymbol{H}/dt$. Our study shows that in addition to substorms, the most intense external $|d\boldsymbol{H}/dt|$ can be caused by other event types as well. These event types are directly driven by solar wind disturbances and current global MHD models work reasonably well in predicting the associated external geomagnetic disturbances. Whichever forecast methods are used, an experienced scientist-in-the-loop is needed for understanding uncertainties. Such experience can only be reached by analysing a large set of events such as in this study.

## 5 Conclusions

We have compiled a list of the most intense external (due to ionospheric and magnetospheric electric currents), internal (due to induced telluric currents), and total (external + internal) $|d\boldsymbol{H}/dt|$ (amplitude of the 10 s time derivative of the horizontal geomagnetic field) events at all IMAGE stations between 1994 and 2018 and examined in detail five events that caused the most intense external $|d\boldsymbol{H}/dt|$ values. Our conclusions are as follows:



1. In agreement with previous results, the most intense external $|d\mathbf{H}/dt|$ were broadly associated with sudden commencements (SCs), pulsations, and substorms during geomagnetic storms preceded by a SC and driven by intense southward IMF and often fast solar wind. Intense GIC peaks were associated with intense external $|d\mathbf{H}/dt|$ nearby.

2. In the SC event, the most intense $|d\mathbf{H}/dt|$ in the prenoon sector was caused by an intensification of the EEJ due to an abrupt compression of the magnetopause.

3. In the pulsation event, the most intense $|d\mathbf{H}/dt|$ in the prenoon sector was caused by disruption of the EEJ due to wave activity during a magnetopause expansion as an intense solar wind dynamic pressure weakened. This event took place during the sheath of an ICME.

4. In the premidnight substorm event, the most intense $|d\mathbf{H}/dt|$ was caused by sudden weakening and poleward retreat of the WEJ associated with wave activity. The WEJ weakening and poleward retreat coincided with an inferred weakening
of dipolarization and, consequently, expansion of the transition region in the magnetosphere.

5. In the two morning sector events associated with substorm activity during ICME clouds, the most intense $|d\mathbf{H}/dt|$ was caused by an intensified, undulating WEJ. In one case, weakening of the WEJ, and in the other, change in the current direction, produced the most intense signature. The northward and southward motion of the entire WEJ indicates corresponding changes in the magnetospheric magnetic field configuration. Fast magnetospheric flows associated with
auroral streamer or omega band activity could be possible drivers.

6. Despite being associated with various event types and occurring at different local time sectors, there were common features among the drivers of most intense $|d\mathbf{H}/dt|$ values: pre-existing intense ionospheric currents (the SC event was an exception) that were abruptly modified by sudden changes in the magnetospheric magnetic field configuration.

7. While the most intense external $|d\mathbf{H}/dt|$ events at adjacent stations typically occurred simultaneously, the most intense
internal and total $|d\mathbf{H}/dt|$ events were more scattered in time, most likely due to the complexity of induction in the conducting ground. From the analysis viewpoint, field separation is useful, since it removes the complicated contribution by induced currents, especially in $|d\mathbf{H}/dt|$.

8. While proper description of the fast changes during SC events appears to require 1 s data, pulsations and substorms may be sufficiently described by 10 s $|d\mathbf{H}/dt|$. 1 min data, however, significantly underestimates the $|d\mathbf{H}/dt|$ peaks.

*Code and data availability.*   IMAGE data are available at https://space.fmi.fi/image. GIC recordings from MAN are available at https://space.fmi.fi/gic. The code used to calculate magnetic coordinates and local times is available at https://apexpy.readthedocs.io/en/latest/. The code used to calculate the wavelet transforms is available at https://pywavelets.readthedocs.io/en/latest/. The list of sudden commencements (SC) was extracted from http://isgi.unistra.fr/. Solar wind, SYM-H, and auroral electrojet index data were extracted from NASA/GSFC's OMNI data set through OMNIWeb https://omniweb.gsfc.nasa.gov/.



*Video supplement.* IMAGE_19990417T025550_10sec_19990417T032550.mp4, IMAGE_20011124T071720_10sec_20011124T074720.mp4, IMAGE_20031029T060650_10sec_20031029T061650.mp4, IMAGE_20031030T194340_10sec_20031030T201840.mp4, and IMAGE_20061215T015210_10sec_20061215T022210.mp4 illustrate the time development of the ionospheric equivalent currents and the magnetic field they produce on the ground on 17 Apr 1999 from 02:55:50 UT to 03:25:50 UT (30 min), on 24 Nov 2001 from 07:17:20 UT to 07:47:20 UT (30 min), on 29 Oct 2003 from 06:06:50 UT to 06:16:50 UT (10 min), on 30 Oct 2003 from 19:43:40 UT to 20:18:40 UT

(35 min), and on 15 Dec 2006 from 01:52:10 UT to 02:22:10 UT (30 min), respectively, with a 10 s time step. The animations consist of frames similar to Fig. 9a–e, 9k, and 9l.

*Author contributions.* L. J. prepared the material and wrote the manuscript together with A. V. A. D. provided expertise on dB/dt and GIC, M. K. on the behavior of internal and external H and dH/dt, A. S. on storm-time dB/dt spikes, and J. W. on omega bands and streamers. All coauthors read the manuscript and commented on it.

*Competing interests.* The authors declare that they have no conflict of interest.

*Acknowledgements.* This work was supported by the Academy of Finland (grant no. 314670 and no. 339329). We thank the institutes that maintain the IMAGE Magnetometer Array: Tromsø Geophysical Observatory of UiT the Arctic University of Norway (Norway), Finnish Meteorological Institute (Finland), Institute of Geophysics Polish Academy of Sciences (Poland), GFZ German Research Centre for Geosciences (Germany), Geological Survey of Sweden (Sweden), Swedish Institute of Space Physics (Sweden), Sodankylä Geophysical Observatory of

480 the University of Oulu (Finland), Polar Geophysical Institute (Russia), and DTU Technical University of Denmark (Denmark). GIC recordings are maintained by the Finnish Meteorological Institute, supported by Gasum Oy until December 2010. In March 2011 to February 2014, maintenance of recordings received funding from the European Community's Seventh Framework Programme (FP7/2007-2013) under grant agreement no 260330 (EURISGIC). We acknowledge use of NASA/GSFC's Space Physics Data Facility's OMNIWeb service, and OMNI data. The results presented in this paper rely on geomagnetic indices calculated and made available by ISGI Collaborating Institutes from

485 data collected at magnetic observatories. We thank the involved national institutes, the INTERMAGNET network and ISGI (isgi.unistra.fr). We thank Elena Marshalko for commenting on the manuscript.





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

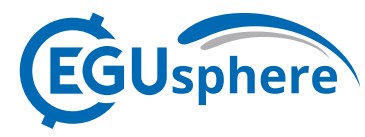

**Table 1.** Maximum total (external + internal), external (ionospheric and magnetospheric), and internal (telluric) time derivative of the horizontal ground magnetic field ($|d\mathbf{H}/dt| = \sqrt{(dB_x/dt)^2 + (dB_y/dt)^2}$ [nTs$^{-1}$]) at IMAGE stations between 1994 and 2018. Magnetic local time (MLT) [HH:MM] has been calculated using https://apexpy.readthedocs.io/en/latest/. SYM-H [nT] has been obtained from https://spdf.gsfc.nasa.gov/pub/data/omni/high_res_omni/. Colors: Event 1: 29–31 Oct 2003, Event 2: 07–08 Sep 2017, and Event 3: 24 Nov 2001. Colored dots in the second column indicate station availability during the days of the Events (black: not available). The absolute maximum total, external, and internal $|d\mathbf{H}/dt|$ has been **highlighted**.

| Station | Total UT | Total MLT | Total \|dH/dt\| | Total SYM-H | External UT | External MLT | External \|dH/dt\| | External SYM-H | Internal UT | Internal MLT | Internal \|dH/dt\| | Internal SYM-H |
|---|---|---|---|---|---|---|---|---|---|---|---|---|
| NAL | 2005 Jan 21 17:22:10 | 20:06 | 39.7 | 55 | 2005 Jan 02 15:49:40 | 18:46 | 12.0 | -35 | 2005 Jan 21 17:22:10 | 20:06 | 33.1 | 55 |
| LYR | 2005 Jan 21 17:22:10 | 20:08 | 46.7 | 55 | 2005 Jan 02 15:49:40 | 18:48 | 17.4 | -35 | 1994 Feb 21 09:01:50$^a$ | 11:57 | 38.1 | 57 |
| HOR | 2003 Mar 17 21:17:10 | 23:58 | 47.2 | -66 | 2005 Jan 02 15:49:40 | 18:38 | 14.0 | -35 | 2003 Mar 17 21:17:10 | 23:58 | 41.2 | -66 |
| HOP | 2003 Oct 30 16:49:30 | 20:14 | 46.7 | -108 | 2005 Jan 02 15:49:40 | 18:58 | 8.3 | -35 | 2003 Oct 30 16:49:30 | 20:14 | 39.9 | -108 |
| BJN | 2003 Oct 21 17:02:10 | 19:58 | 42.4 | -61 | 1995 Oct 04 15:01:30 | 17:54 | 7.7 | -83 | 2003 Oct 21 17:02:10 | 19:58 | 35.6 | -61 |
| NOR | 2017 Sep 12 20:44:10 | 23:37 | 20.7 | 15 | 2015 Oct 08 18:04:50 | 21:01 | 13.2 | -50 | 2016 Dec 23 18:17:00 | 20:56 | 15.8 | -40 |
| SOR | 2003 Oct 29 06:12:00$^a$ | 09:07 | 32.6 | 41 | 2001 Nov 24 07:32:20 | 10:33 | 18.2 | -97 | 2005 Feb 18 01:06:20 | 03:14 | 24.4 | -85 |
| TRO | 2001 Nov 24 07:32:20 | 10:20 | 33.6 | -97 | 2001 Nov 24 07:32:20 | 10:20 | 19.2 | -97 | 1995 Jan 30 21:05:50 | 23:00 | 27.3 | -29 |
| AND | 2003 Oct 29 06:12:00$^a$ | 08:44 | 36.0 | 41 | 2001 Nov 24 07:32:20 | 10:10 | 17.8 | -97 | 2013 Jul 06 02:59:30 | 05:10 | 26.9 | -37 |
| KEV | 2003 Oct 29 06:11:30$^a$ | 09:17 | 30.5 | -18 | 2003 Oct 29 06:11:50$^a$ | 09:18 | 16.5 | 41 | 2003 Oct 29 06:11:30$^a$ | 09:17 | 35.0 | -18 |
| MAS | 2001 Nov 24 07:32:20 | 10:33 | 43.7 | -97 | 2003 Oct 29 06:11:50$^a$ | 09:07 | 20.3 | 41 | 2001 Nov 24 07:32:20 | 10:33 | 25.9 | -97 |
| KIL | 2003 Oct 29 06:11:50$^a$ | 08:57 | 42.8 | 41 | 2003 Oct 29 06:11:50$^a$ | 08:57 | 20.8 | 41 | 2003 Oct 29 06:11:50$^a$ | 08:57 | 22.3 | 41 |
| LEK | 2003 Oct 29 06:11:50$^a$ | 08:33 | 31.7 | 41 | 2001 Nov 24 07:32:20 | 09:58 | 14.9 | -97 | 2002 Oct 04 18:09:30 | 20:18 | 20.2 | -63 |
| ABK | 1999 Apr 17 03:10:40 | 05:26 | 36.8 | -100 | 2003 Oct 29 06:11:50$^a$ | 08:49 | 20.3 | 41 | 1999 Apr 17 03:10:40 | 05:26 | 19.8 | -100 |
| IVA | 2003 Oct 29 06:11:40$^a$ | 09:15 | 43.2 | 41 | 2003 Oct 29 06:11:40$^a$ | 09:15 | 15.0 | 41 | 2003 Oct 29 06:11:40$^a$ | 09:15 | 28.2 | 41 |
| KIR | 2006 Dec 15 02:07:10 | 04:14 | 37.8 | -174 | **2006 Dec 15 02:07:10** | 04:14 | **21.2** | -174 | 2006 Dec 15 02:07:10 | 04:14 | 18.3 | -174 |
| MUO | 2003 Oct 29 06:11:50$^a$ | 09:02 | 37.6 | 41 | 2003 Oct 29 06:11:50$^a$ | 09:02 | 18.8 | 41 | 1998 May 04 04:01:30 | 06:33 | 23.5 | -187 |
| SOD | 2017 Sep 08 00:03:20 | 02:42 | 33.1 | -94 | 2017 Sep 07 23:15:40 | 01:56 | 13.2 | -19 | 2017 Sep 08 00:24:40 | 03:03 | 22.4 | -108 |
| DON | 2017 Sep 08 00:30:00 | 02:22 | 27.8 | -120 | 2015 Sep 08 01:15:00 | 03:06 | 9.1 | -80 | 2017 Sep 08 00:30:00 | 02:22 | 22.6 | -120 |
| JCK | 2012 Oct 09 01:13:00 | 03:22 | 30.7 | -93 | 2017 Sep 08 00:33:20 | 02:40 | 11.3 | -132 | 2012 Oct 09 01:13:00 | 03:22 | 24.6 | -93 |
| PEL | 2003 Oct 29 06:11:40$^a$ | 09:00 | 52.7 | 41 | 2003 Oct 29 06:11:40$^a$ | 09:00 | 20.1 | 41 | 2003 Oct 29 06:11:40$^a$ | 09:00 | 32.6 | 41 |
| RVK | **2003 Oct 29 06:11:40$^a$** | **08:15** | **77.0** | 41 | 2003 Oct 29 06:11:40$^a$ | 08:15 | 16.2 | 41 | **2003 Oct 29 06:11:40$^a$** | **08:15** | **61.3** | 41 |
| RAN | 2017 Sep 08 00:03:40 | 02:38 | 33.6 | -96 | 2017 Sep 07 23:15:50 | 01:51 | 11.8 | -19 | 2017 Sep 08 00:03:40 | 02:38 | 26.5 | -96 |
| LYC | 2015 Mar 17 17:33:50 | 19:28 | 31.2 | -165 | 2000 Nov 10 06:28:40$^a$ | 08:57 | 8.0 | 24 | 2017 Sep 08 00:27:10 | 02:34 | 25.0 | -117 |
| OUJ | 2003 Oct 30 20:08:40 | 22:46 | 32.2 | -187 | 2003 Oct 30 20:08:40 | 22:46 | 15.6 | -187 | 2017 Sep 08 00:33:20 | 03:06 | 24.9 | -132 |
| DOB | 2003 Oct 29 06:11:40$^a$ | 08:01 | 24.8 | 41 | 2003 Oct 29 06:11:40$^a$ | 08:01 | 13.3 | 41 | 2004 Nov 09 19:45:20 | 21:20 | 16.0 | -142 |
| SOL | 2017 Sep 08 00:28:50 | 01:44 | 18.7 | -118 | 2017 Sep 08 17:56:00 | 19:13 | 6.7 | -106 | 2017 Sep 08 00:28:50 | 01:44 | 12.9 | -118 |
| MEK | 2012 Mar 15 17:01:10 | 19:29 | 14.0 | -54 | 2017 Sep 07 23:15:50 | 01:58 | 6.0 | -19 | 2017 Sep 08 00:31:10 | 03:12 | 11.5 | -125 |
| HAN | 2003 Oct 29 07:27:20 | 10:15 | 41.6 | -92 | 2003 Oct 30 20:08:50 | 22:39 | 14.0 | -187 | 2003 Oct 29 07:27:20 | 10:15 | 35.1 | -92 |
| KAR | 2004 Nov 09 19:48:20 | 21:04 | 13.3 | -142 | 2017 Sep 08 17:57:50 | 19:12 | 4.5 | -105 | 2004 Nov 09 19:45:40 | 21:02 | 10.7 | -145 |
| NUR | 2003 Oct 30 21:17:30 | 23:36 | 28.3 | -302 | 2003 Oct 30 20:08:40 | 22:30 | 11.0 | -187 | 2003 Oct 30 21:17:30 | 23:36 | 21.0 | -302 |
| UPS | 2003 Oct 30 20:08:50 | 22:05 | 19.4 | -187 | 2003 Oct 30 20:08:40 | 22:05 | 10.3 | -187 | 2003 Oct 29 07:03:20 | 09:16 | 12.4 | -39 |
| TAR | 2015 Mar 17 17:33:50 | 19:40 | 16.7 | -165 | 2015 Mar 17 17:33:50 | 19:40 | 5.8 | -165 | 2015 Mar 17 17:33:50 | 19:40 | 10.9 | -165 |
| BRZ | 2015 Mar 17 17:35:30 | 19:33 | 8.0 | -164 | 2015 Mar 17 17:35:30 | 19:33 | 2.9 | -164 | 2015 Mar 17 17:35:30 | 19:33 | 5.5 | -164 |
| HLP | 2018 Aug 20 10:23:00 | 11:53 | 1.4 | -8 | 2018 Aug 20 10:23:00 | 11:53 | 0.6 | -8 | 2018 Aug 20 10:23:00 | 11:53 | 0.8 | -8 |
| WNG | 2018 Dec 28 10:14:40 | 11:44 | 1.5 | -10 | 2017 Sep 08 17:56:00 | 19:12 | 1.7 | -106 | 2017 Sep 08 17:56:00 | 19:12 | 1.9 | -106 |
| SUW | 2017 Nov 30 06:25:40 | 08:45 | 2.2 | -3 | 2015 Dec 20 16:31:00 | 18:31 | 0.7 | -94 | 2017 Nov 30 06:25:40 | 08:45 | 1.9 | -3 |
| NGK | 2017 Jul 16 06:01:30$^a$ | 07:11 | 1.5 | 26 | 2017 Sep 08 17:56:00 | 19:21 | 0.8 | -106 | 2017 Sep 08 17:56:10 | 19:21 | 1.1 | -106 |
| PPN | 2018 Aug 20 10:22:50 | 12:04 | 1.2 | -8 | 2018 Aug 20 10:22:50 | 12:04 | 0.5 | -8 | 2018 Aug 27 15:19:40 | 17:07 | 0.8 | -64 |

$^a$SC according to http://isgi.unistra.fr/





**Table 2.** The same as Table 1 except for $|\boldsymbol{H}|$ instead of $|d\boldsymbol{H}/dt|$.

| Station | Available | Measured | | | | External | | | | Internal | | | |
|---|---|---|---|---|---|---|---|---|---|---|---|---|---|
| | | UT | MLT | $\|\boldsymbol{H}\|$ | SYM-H | UT | MLT | $\|\boldsymbol{H}\|$ | SYM-H | UT | MLT | $\|\boldsymbol{H}\|$ | SYM-H |
| NAL | | 2005 Jan 07 23:07:00 | 01:38 | 2430.1 | -82 | 2003 Oct 30 20:07:30 | 23:14 | 997.3 | -132 | 2005 Jan 07 23:07:00 | 01:38 | 1894.8 | -82 |
| LYR | | 2005 Jan 07 23:08:10 | 01:41 | 2864.9 | -85 | 2005 Jan 07 23:08:00 | 01:41 | 1199.3 | -85 | 2005 Jan 07 23:08:10 | 01:41 | 1832.4 | -85 |
| HOR | | 2005 Jan 07 23:08:20 | 01:31 | 2928.8 | -85 | 2005 Jan 07 23:08:10 | 01:31 | 1366.3 | -85 | 2005 Jan 07 23:08:20 | 01:31 | 1876.4 | -85 |
| HOP | | 2004 Nov 08 01:27:40 | 04:41 | 3119.7 | -192 | 2004 Nov 08 01:56:30 | 05:11 | 1097.2 | -231 | 2004 Nov 08 01:21:10 | 04:34 | 2419.8 | -174 |
| BJN | | 2005 Jan 07 22:46:20 | 01:03 | 2684.3 | -43 | 2005 Jan 07 23:10:20 | 01:26 | 1117.5 | -82 | 2005 Jan 07 22:46:20 | 01:03 | 1691.0 | -43 |
| NOR | | 2017 Sep 08 00:17:30 | 03:06 | 1963.1 | -97 | 2017 Sep 08 00:17:50 | 03:06 | 1521.0 | -100 | 2017 Sep 08 15:52:50 | 18:37 | 877.9 | -104 |
| SOR | | 2003 Oct 30 20:10:10 | 22:51 | 3018.9 | -157 | 2003 Oct 30 19:55:00 | 22:36 | 2238.3 | -88 | 2004 Nov 08 02:04:00 | 04:41 | 1213.4 | -234 |
| TRO | | 2003 Oct 30 19:54:50 | 22:23 | 2950.6 | -88 | 2003 Oct 30 19:55:00 | 22:23 | 2370.6 | -88 | 2017 Sep 08 00:16:00 | 02:39 | 1181.3 | -96 |
| AND | | 2003 Oct 30 19:56:40 | 22:15 | 3311.3 | -109 | 2003 Oct 30 19:55:00 | 22:14 | 2190.2 | -88 | 2017 Sep 08 00:15:10 | 02:29 | 1848.0 | -96 |
| KEV | | 2003 Oct 30 19:56:30 | 22:49 | 3900.8 | -87 | 2003 Oct 30 19:56:30 | 22:49 | 2753.4 | -87 | 2003 Oct 30 19:56:30 | 22:49 | 1168.6 | -87 |
| MAS | | 2003 Oct 29 22:42:00 | 01:19 | 2630.5 | -279 | 2003 Oct 29 06:39:00 | 09:36 | 1835.4 | -7 | 2003 Oct 29 22:40:50 | 01:18 | 1034.7 | -284 |
| KIL | | 2003 Oct 30 19:55:00 | 22:27 | 3605.3 | -88 | 2003 Oct 30 19:56:10 | 22:28 | 2814.7 | -87 | 2003 Oct 30 20:07:10 | 22:38 | 1102.4 | -132 |
| LEK | | 2003 Oct 30 20:07:10 | 22:14 | 3060.6 | -132 | 2003 Oct 30 19:56:10 | 22:03 | 2308.0 | -87 | 2003 May 10 02:49:20 | 04:55 | 1043.4 | -81 |
| ABK | | 2003 Oct 30 20:07:30 | 22:30 | 3831.7 | -132 | 2003 Oct 30 19:56:10 | 22:20 | 2863.9 | -87 | 2003 Oct 30 20:07:30 | 22:30 | 1249.4 | -132 |
| IVA | | 2003 Oct 30 20:08:10 | 22:57 | 3934.5 | -75 | 2003 Oct 30 19:56:30 | 22:46 | 2789.1 | -87 | 2003 Oct 30 20:08:10 | 22:57 | 1279.6 | -75 |
| KIR | | 2017 Sep 08 00:18:30 | 02:40 | 3319.3 | -100 | 2017 Sep 08 00:18:10 | 02:40 | 2483.0 | -100 | 2017 Sep 08 00:19:00 | 02:40 | 999.9 | -103 |
| MUO | | 2003 Oct 30 19:56:30 | 22:33 | 3655.9 | -87 | 2003 Oct 30 19:56:20 | 22:33 | 3106.4 | -87 | 2003 Oct 30 19:54:40 | 22:31 | 961.3 | -88 |
| SOD | | 2017 Sep 08 00:21:50 | 03:00 | 3745.0 | -105 | 2017 Sep 08 00:21:50 | 03:00 | 2358.1 | -105 | 2017 Sep 08 00:21:50 | 03:00 | 1425.4 | -105 |
| DON | | 2017 Sep 08 00:18:00 | 02:10 | 3677.1 | -100 | 2017 Sep 08 00:18:40 | 02:11 | 2159.9 | -103 | 2017 Sep 08 00:17:50 | 02:10 | 1543.0 | -100 |
| JCK | | 2017 Sep 08 00:19:10 | 02:26 | 3710.6 | -103 | 2017 Sep 08 00:18:50 | 02:25 | 2280.1 | -103 | 2017 Sep 08 00:19:20 | 02:26 | 1477.1 | -103 |
| PEL | | 2003 Oct 30 19:56:30 | 22:31 | 4210.0 | -87 | 2003 Oct 30 20:08:10 | 22:42 | 2863.9 | -75 | 2017 Sep 08 00:20:50 | 02:50 | 1486.0 | -107 |
| RVK | | 2003 Oct 30 20:07:30 | 21:57 | 3901.7 | -132 | 2003 Oct 30 20:07:30 | 21:57 | 2426.9 | -132 | 2017 Sep 08 00:23:40 | 02:08 | 1941.9 | -106 |
| RAN | | 2017 Sep 08 00:25:10 | 02:59 | 4026.6 | -108 | 2017 Sep 08 00:24:40 | 02:58 | 2310.8 | -108 | 2017 Sep 08 00:25:20 | 02:59 | 1810.6 | -108 |
| LYC | | 2017 Sep 08 00:24:00 | 02:31 | 3794.2 | -106 | 2017 Sep 08 00:24:00 | 02:31 | 1960.1 | -106 | 2017 Sep 08 00:25:30 | 02:32 | 1863.7 | -108 |
| OUJ | | 2003 Oct 30 20:05:50 | 22:44 | 4719.2 | -146 | 2003 Oct 30 20:05:50 | 22:44 | 2568.1 | -146 | 2003 Oct 30 20:05:50 | 22:44 | 2342.5 | -146 |
| DOB | | 2003 Oct 30 20:07:30 | 21:43 | 2967.9 | -132 | 2003 Oct 30 20:08:10 | 21:44 | 1905.1 | -75 | 2003 Oct 29 06:47:40 | 08:39 | 1305.5 | 10 |
| SOL | | 2017 Sep 08 00:28:10 | 01:44 | 1820.1 | -118 | 2017 Sep 08 00:34:20 | 01:50 | 910.6 | -133 | 2017 Sep 08 00:28:10 | 01:44 | 1055.6 | -118 |
| MEK | | 2004 Nov 07 23:53:30 | 02:32 | 2084.8 | -194 | 2004 Nov 07 23:52:40 | 02:31 | 1174.0 | -194 | 2017 Sep 08 00:33:10 | 03:14 | 961.7 | -132 |
| HAN | | 2003 Oct 30 20:04:10 | 22:35 | 3738.8 | -139 | 2003 Oct 30 20:04:10 | 22:35 | 2214.6 | -139 | 2003 Oct 30 20:04:20 | 22:35 | 1731.2 | -139 |
| KAR | | 2004 Nov 07 23:53:10 | 01:03 | 1726.2 | -194 | 2004 Nov 07 23:53:10 | 01:03 | 837.5 | -194 | 2004 Nov 07 23:53:10 | 01:03 | 891.5 | -194 |
| NUR | | 2003 Oct 30 20:04:20 | 22:25 | 2949.5 | -139 | 2003 Oct 30 20:04:20 | 22:25 | 1650.6 | -139 | 2003 Oct 30 21:18:50 | 23:37 | 1470.4 | -292 |
| UPS | | 2003 Oct 30 20:04:00 | 22:01 | 2743.2 | -139 | 2003 Oct 30 20:04:00 | 22:01 | 1547.0 | -139 | 2003 Oct 30 20:04:10 | 22:01 | 1204.5 | -139 |
| TAR | | 2003 Oct 29 22:33:00 | 00:52 | 2002.4 | -285 | 2001 Nov 06 03:01:20 | 05:25 | 788.9 | -261 | 2003 Oct 29 22:33:10 | 00:52 | 1337.6 | -285 |
| BRZ | | 2015 Mar 17 17:38:40 | 19:36 | 579.3 | -164 | 2015 Mar 17 23:14:20 | 01:10 | 388.1 | -213 | 2015 Mar 17 17:39:00 | 19:37 | 326.0 | -164 |
| HLP | | 2018 Aug 26 05:41:20 | 07:22 | 152.3 | -181 | 2018 Aug 26 03:20:50 | 05:04 | 111.2 | -118 | 2018 Aug 26 05:37:20 | 07:18 | 96.7 | -177 |
| WNG | | 2017 Sep 07 23:31:50 | 00:48 | 262.8 | -90 | 2017 Sep 07 23:30:10 | 00:46 | 208.9 | -78 | 2017 Sep 08 00:26:50 | 01:42 | 230.3 | -117 |
| SUW | | 2015 Dec 20 19:06:40 | 20:57 | 220.9 | -119 | 2017 Sep 08 00:14:30 | 02:16 | 142.1 | -97 | 2017 Sep 08 00:26:40 | 02:28 | 170.2 | -117 |
| NGK | | 2017 Sep 07 23:29:40 | 00:55 | 238.9 | -78 | 2017 Sep 07 23:30:10 | 00:56 | 184.8 | -78 | 2017 Sep 08 00:26:50 | 01:51 | 183.3 | -117 |
| PPN | | 2018 Aug 26 07:06:20 | 08:55 | 143.4 | -200 | 2018 Aug 26 03:19:40 | 05:13 | 105.9 | -118 | 2018 Aug 26 07:07:00 | 08:56 | 98.3 | -201 |





**Figure 10.** (a): Interplanetary magnetic field (IMF) magnitude and GSM components ±1 h around the event on 24 Nov 2001 at 07:32:20 UT. (b): GSM x component of the solar wind velocity. (c): Solar wind density. (d): Solar wind dynamic pressure. (e): SYM-H index. (f): Auroral electrojet indices. The time interval of Fig. 11 (30 min) is shaded in grey and the time of the extreme $|d\boldsymbol{H}/dt|$ event at IMAGE is indicated by the black dashed vertical line.





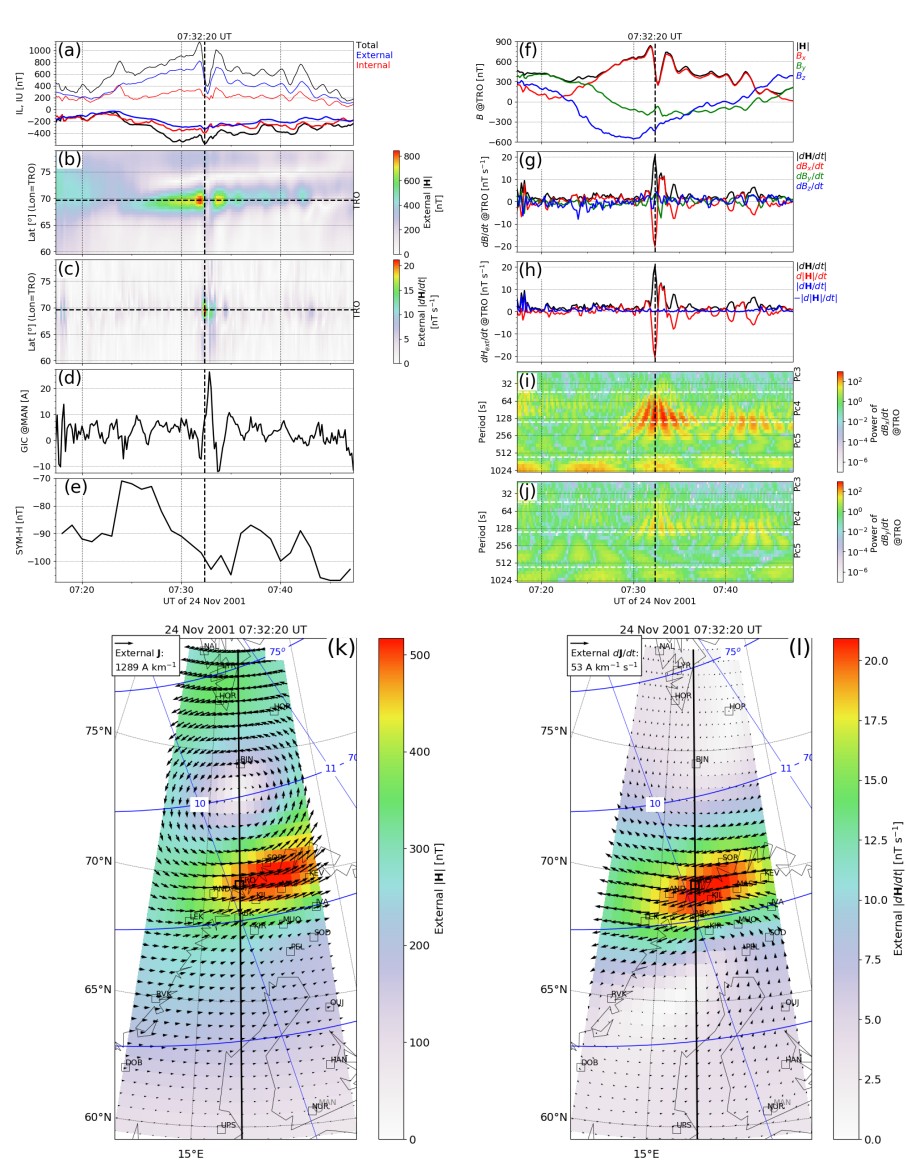

**Figure 11.** Data in a format similar to Fig. 9 ±15 min around the event on 24 Nov 2001 at 07:32:20 UT.





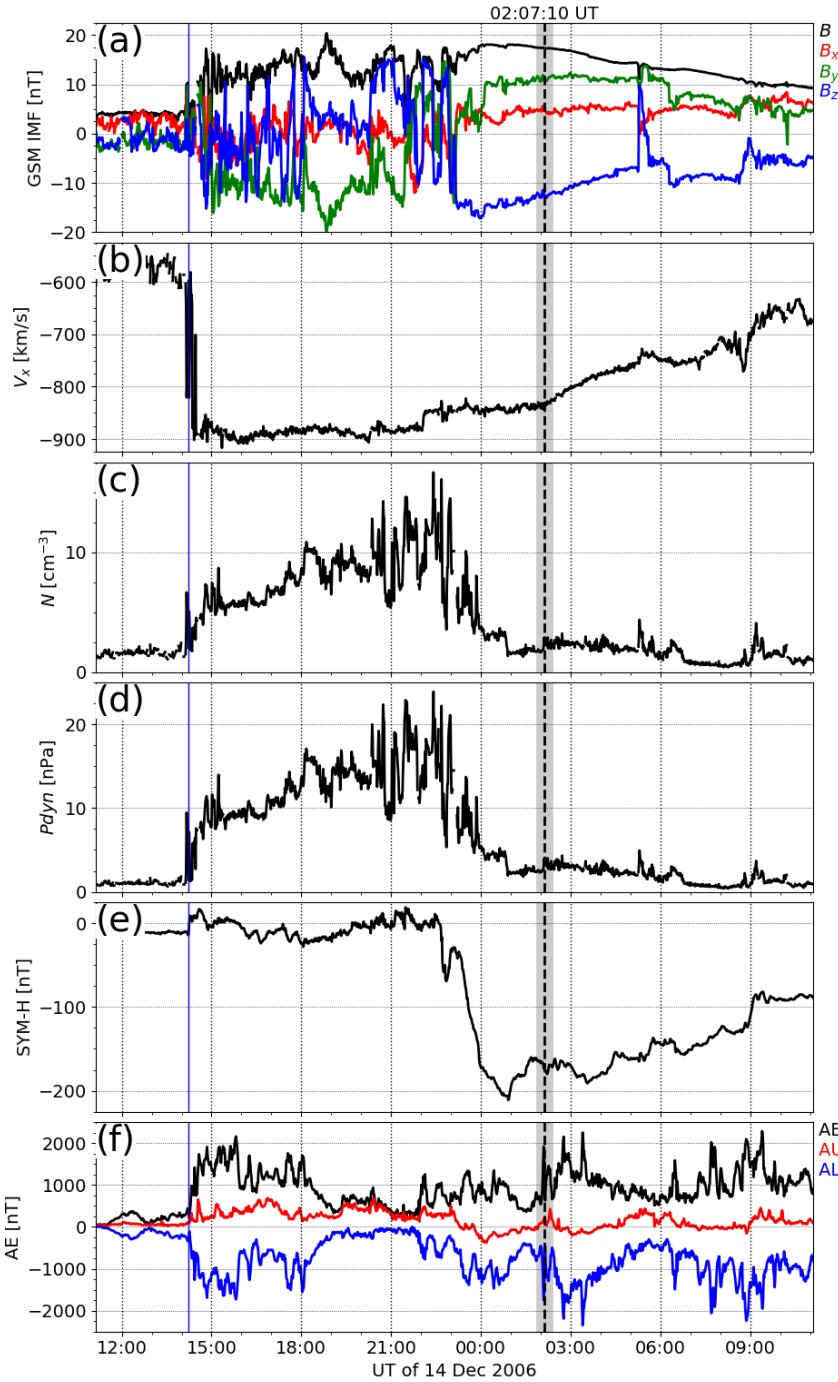

**Figure 12.** Data in a format similar to Fig. 10 from −15 h to +9 h around the event on 15 Dec 2006 at 02:07:10 UT. The vertical blue line indicates a SC at 14:14:18 UT.





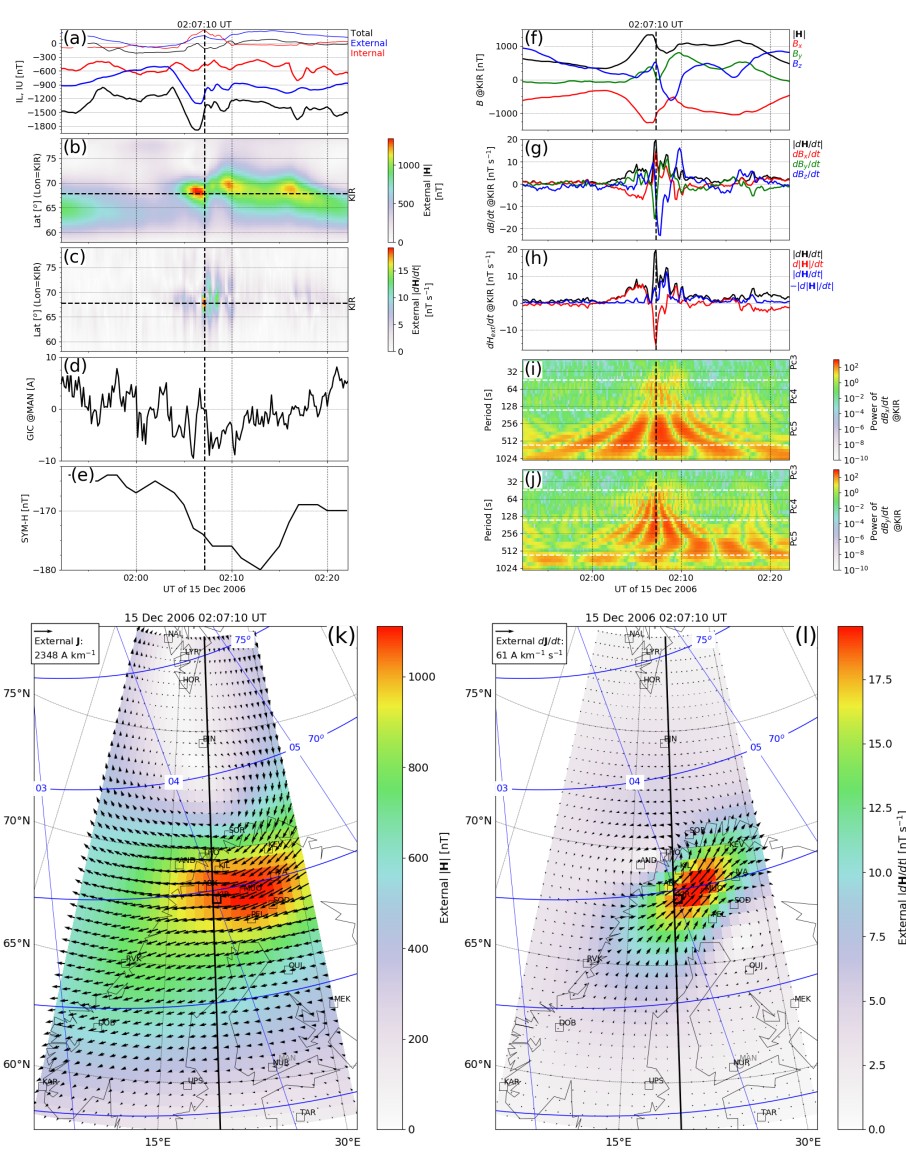

**Figure 13.** Data in a format similar to Fig. 9 ±15 min around the event on 15 Dec 2006 at 02:07:10 UT.





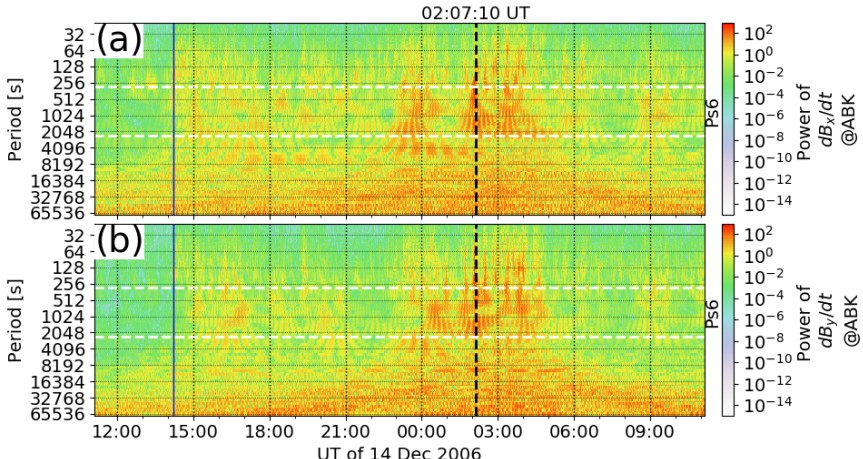

**Figure 14.** Wavelet tranform of external $dB_x/dt$ and $dB_y/dt$ at KIR for the longer time interval from $-15$ h to $+9$ h of Fig. 13 compared to the $\pm15$ min shown in Fig. 13i–j for the event on 15 Dec 2006 at 02:07:10 UT. The period range of the ultra-low frequency (ULF) pulsation class Ps6 (5–40 min) (Jacobs et al., 1964) are shown with the white, horizontal, dashed lines. The vertical solid blue line indicates the SC at 14:14:18 UT and the vertical dashed black line indicates the time of the extreme external $|d\mathbf{H}/dt|$ observed at KIR.

**Table 3.** Summary of the five studied events of the most intense external 10 s $|d\mathbf{H}/dt|$ as observed by IMAGE between 1994 and 2018.

| UT | MLT [h] | $|d\mathbf{H}/dt|$ [nTs$^{-1}$] | Event type | Ionospheric currents | Possible driver |
|---|---|---|---|---|---|
| 2006 Dec 15 02:07:10 | 04 | 21.2 | Substorm | Undulating, intensified WEJ | Magnetospheric fast flow? |
| 2003 Oct 29 06:11:40–50 | 09 | 20.8 | SC | EEJ intensification & poleward motion | Magnetopause compression |
| 2001 Nov 24 07:32:20 | 10 | 19.2 | Pulsation | EEJ disruption by ULF waves | Magnetopause expansion |
| 1999 Apr 04 03:10:50 | 05 | 18.0 | Substorm | Undulating, intensified WEJ | Magnetospheric fast flow? |
| 2003 Oct 30 20:08:40 | 23 | 15.6 | Substorm | WEJ weakening & poleward retreat, ULF waves | Transition region expansion |



**Figure 15.** Data in a format similar to Fig. 10 ±20 h around the event on 17 Apr 1999 at 03:10:50 UT. The vertical blue lines indicates a SC at 11:24:54 UT.



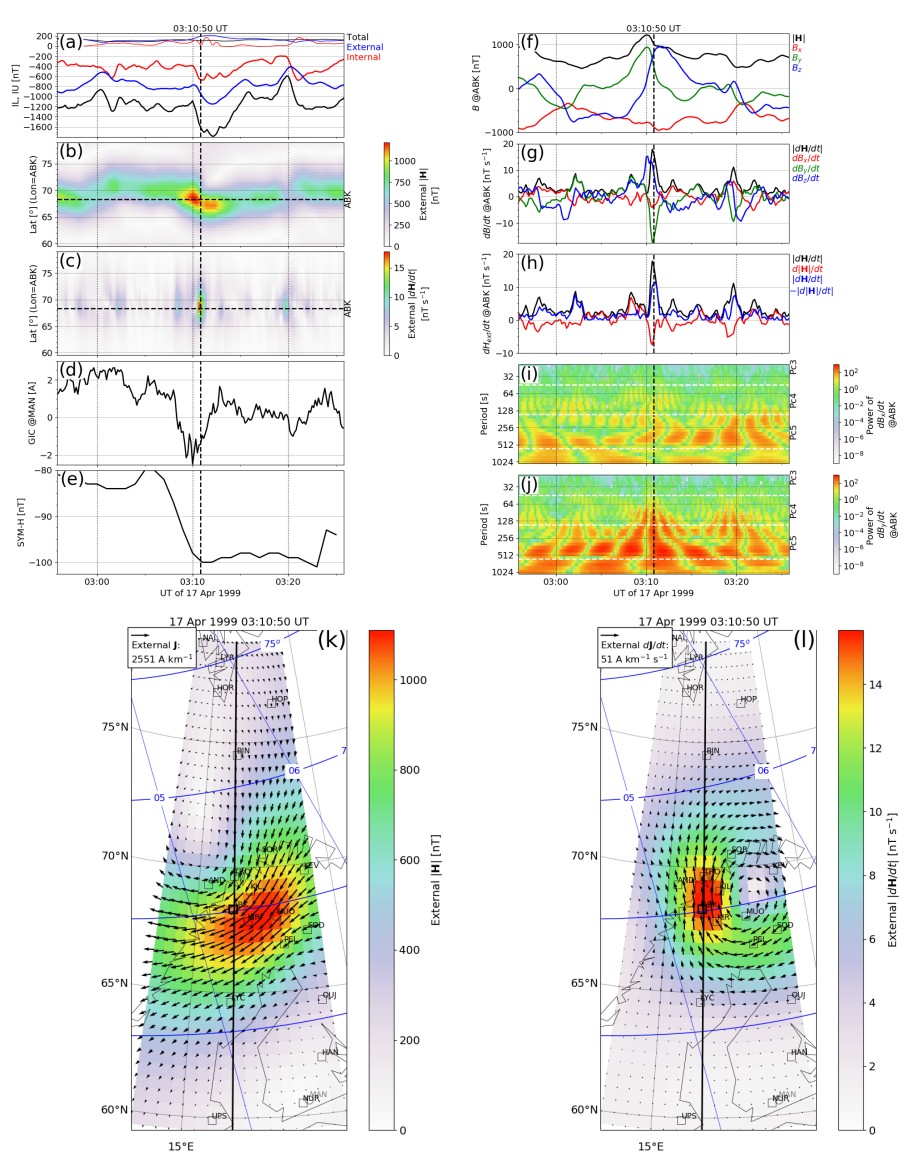

**Figure 16.** Data in a format similar to Fig. 9 ±15 min around the event on 17 Apr 1999 at 03:10:50 UT.




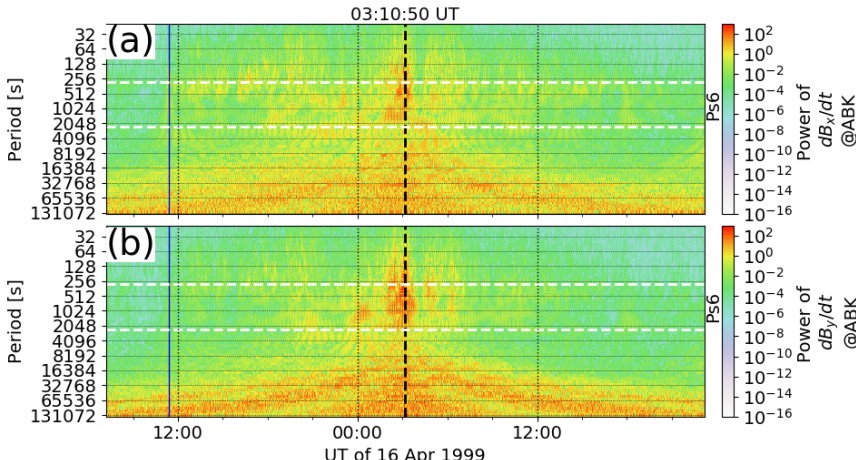

**Figure 17.** Data in a format similar to Fig. 14 ±20 h around the event on 17 Apr 1999 at 03:10:50 UT.

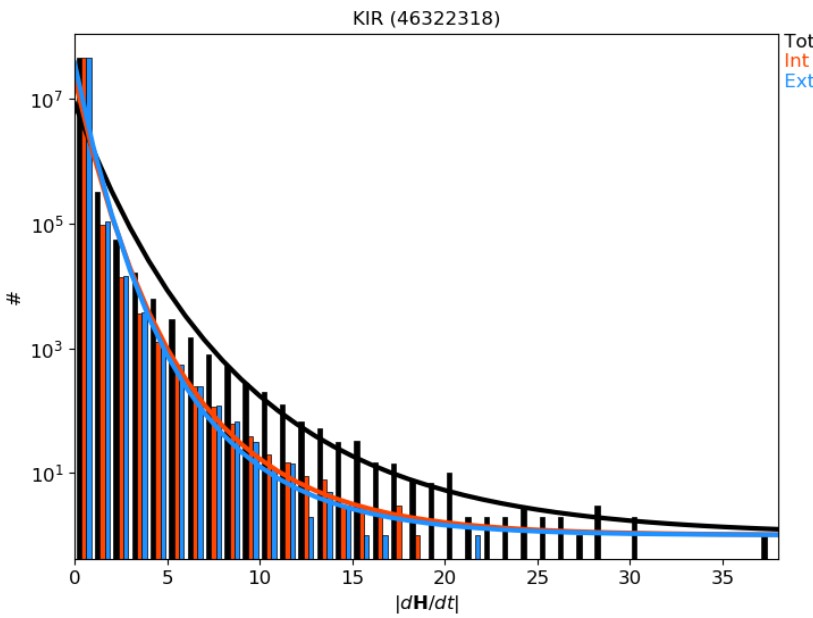

**Figure 18.** Histogram of all total, internal, and external $|d\boldsymbol{H}/dt|$ values observed by KIR between 1994 and 2018. The total number of data points is indicated in the parenthesis. The curves show $a\exp^{-b|d\boldsymbol{H}/dt|}$ function fits to the data points, where $a$ and $b$ are constants.