# Peer review of "Drivers of rapid geomagnetic variations at high latitudes"

_EGUsphere, 2022_

## Author Comment (AC1)

**Reply to Referee**

Liisa Juusola[1], Ari Viljanen[1], Andrew P. Dimmock[2], Mirjam Kellinsalmi[1], Audrey Schillings[3], and James M. Weygand[4]

[1]Finnish Meteorological Institute, Helsinki, Finland
[2]Swedish Institute of Space Physics, Uppsala, Sweden
[3]Department of Physics, Umeå University, Umeå, Sweden
[4]Department of Earth, Planetary, and Space Sciences, University of California Los Angeles, Los Angeles, CA, USA

**Dear Professor Engebretson,**

thank you for a positive and constructive review of our study (https://doi.org/10.5194/egusphere-2022-850). We are happy to carry out all the suggested corrections. Please see below for detailed replies to all comments. The original review is written in black and our replies in blue.

5   **Review of Juusola et al., Drivers of rapid geomagnetic variations at high latitudes, submitted to EGUSphere, 2022**

**General Comments**

This is a very well written study of five of the strongest geomagnetic variations observed by the IMAGE magnetometer array. It has a very good introductory review section, followed by tables showing the largest $|\Delta \boldsymbol{H}|$ and $|d\boldsymbol{H}/dt|$ at each of the IMAGE sites, separated into total (observed) and external and internal contributions. This is followed by a detailed analysis

10  of five events that produced some of the most intense external $|d\boldsymbol{H}/dt|$ values. The authors provide plausible interpretations for the magnetospheric/ionospheric phenomena that drove these events, and also provide a careful discussion in section 4 of some of the limitations of this study (even though it is based on a large volume of data) and of continuing challenges to the successful prediction of intense (dangerous) $|d\boldsymbol{H}/dt|$ events. It concludes that the relevant scientific community is still far from a full understanding of the detailed physical pathway(s) leading to either modest or extreme $|d\boldsymbol{H}/dt|$ events, much less

15  to the prediction of the time and place where these events will occur.

    The content of this paper is of high quality and is certainly appropriate for publication in EGUsphere. This reviewer has only two substantive comments and four more minor comments.

**Specific Comments**

    It is strongly suggested that throughout the paper the magnitude of the perturbations in the horizontal magnetic field

20  that are denoted $|\boldsymbol{H}|$ should be replaced by $|\Delta \boldsymbol{H}|$. The magnitude of the total magnetic field or even its horizontal component (in a given coordinate system) is not what is physically important; it is rather the change in its value (during some appropriate time interval).

Re: OK

    Lines 375–391: The manuscript cites a study by Viljanen et al. (2006) that showed peaks in occurrences of large $|d\boldsymbol{H}/dt|$

25  events 5 minutes after both non-storm and storm-time substorm onsets at Sodankylä (63.92° MLAT) and Nurmijärvi

(56.89° MLAT) during 1997 and 1999 (their Figure 3). However, there were no substorm onsets or sudden intensifications of the western electrojet during the five selected events. The authors may wish to contrast the observations of Viljanen et al. (2006) with those of Engebretson et al. (2021), who showed in their Figure 2 plots of maximum $dB/dt$ events (all $> 6$ nT/s) vs. time delay after substorm onsets for five stations in Arctic Canada during 2015 and 2017, with MLATs ranging from 75.2° to 64.7°. There was no significant peak near 5 minutes after onset at any of these stations (the distributions were relatively flat during the first 30 minutes). The distribution at each station had a gradual and extended falloff that was roughly consistent with those shown in most panels of Figure 3 of Viljanen et al. (2006). The Engebretson et al. (2021) study also showed in panels B and C their Figure 11 that postmidnight dB/dt events that occurred greater than 30 minutes after substorm onsets at the lowest latitude station (KJPK, 64.7° MLAT) occurred during periods of gradual increases in the SML index (weakenings of the WEJ).

Engebretson, M. J., Ahmed, L. Y., Pilipenko, V. A., Steinmetz, E. S., Moldwin, M. B., Connors, M. G., et al. (2021). Superposed epoch analysis of nighttime magnetic perturbation events observed in Arctic Canada. Journal of Geophysical Research: Space Physics, 126, e2021JA029465. https://doi.org/10.1029/2021JA029465

Re: Thank you for reminding us of these results. We suggest to add after line 391: "The second alternative agrees with the results of Viljanen et al. (2006) and Engebretson et al. (2021). Viljanen et al. (2006) examined the occurrence of maximum $|d\mathbf{H}/dt|$ after substorm onsets at IMAGE stations during 1997 and 1999. They showed that the largest value of $|d\mathbf{H}/dt|$ during substorms occurs most probably at about 5 min after the onset at stations with CGM latitude less than 72°. At this time, the amplitude of the westward electrojet increases rapidly. Engebretson et al. (2021) showed the occurrence of maximum $dB/dt$ events vs. time delay after substorm onset for five stations in Arctic Canada during 2015 and 2017, with MLATs ranging from 75.2° to 64.7°. There was no significant peak near 5 min after onset at any of these stations, and it was suggested that maximum $dB/dt$ events are not typically associated with substorm onsets but times of the most intense westward electrojet. The key difference between these apparently contradictory results is that, whereas Viljanen et al. (2006) examined the occurrence of maximum $|d\mathbf{H}/dt|$ after all identified substorm onsets (with average maximum $|d\mathbf{H}/dt|$ typically less than 2.5 nT/s), Engebretson et al. (2021) only considered intense $dB/dt$ events with maximum $dB/dt > 6$ nT/s. Thus, although the intensifying westward electrojet after substorm onset may be a typical source of moderate $|d\mathbf{H}/dt|$ values (Viljanen et al., 2006), the rarer events with strong $dB/dt > 6$ nT/s tend to occur during times of the most intense westward electrojet (Engebretson et al., 2021). Engebretson et al. (2021) also showed that postmidnight $dB/dt$ events that occurred greater than 30 min after substorm onsets at the lowest latitude station (KJPK, 64.7° MLAT) occurred during periods of gradual weakenings of the westward electrojet. These could be similar to our events with the undulating westward electrojet."

**Technical Corrections**

In line 185, the phrase "optimal temporal development" does not seem appropriate. "Optimal" approximates to "best", so this part of the sentence is confusing.

Re: We suggest to remove "optimal".

Figure 4 needs to be much larger in the final published paper, and some of the fine print in the figure could be moved to the figure caption. Figures 5, 6, 9, 11, 13, 14, 16, and 17 would also be easier to read if they made use of the full width of the available space on a page.

Re: We suggest to make a new Fig. 4 (attached Figure 1), which only covers the relevant period 1994–2018, only includes the relevant stations, and does not include any of the fine print or mention the EISCAT magnetometers. The size of the final figures is probably decided by the copyeditors, but we will try to ensure that they are large enough in case the paper is accepted.

In line 285, "possible" should be changed to "possibly."

Re: OK

[Figure]

**Figure 1.** Availability of data from IMAGE magnetometer data.

In line 375, remove "of" after "five."

70      Re: OK

**References**

Engebretson, M. J., Ahmed, L. Y., Pilipenko, V. A., Steinmetz, E. S., Moldwin, M. B., Connors, M. G., Boteler, D. H., Weygand, J. M., Coyle, S., Ohtani, S., Gjerloev, J., and Russell, C. T.: Superposed epoch analysis of nighttime magnetic perturbation events observed in Arctic Canada, J. Geophys. Res.: Space Physics, 126, e2021JA029 465, https://doi.org/https://doi.org/10.1029/2021JA029465, 2021.

75  Viljanen, A., Tanskanen, E. I., and Pulkkinen, A.: Relation between substorm characteristics and rapid temporal variations of the ground magnetic field, Ann. Geophys., 24, 725–733, https://doi.org/https://doi.org/10.5194/angeo-24-725-2006, 2006.

---

## Author Comment (AC3)

**Reply to Referee**

Liisa Juusola[1], Ari Viljanen[1], Andrew P. Dimmock[2], Mirjam Kellinsalmi[1], Audrey Schillings[3], and James M. Weygand[4]

[1]Finnish Meteorological Institute, Helsinki, Finland
[2]Swedish Institute of Space Physics, Uppsala, Sweden
[3]Department of Physics, Umeå University, Umeå, Sweden
[4]Department of Earth, Planetary, and Space Sciences, University of California Los Angeles, Los Angeles, CA, USA

**Dear Reviewer,**

thank you for a constructive and thorough review. We are happy to carry most of the suggested corrections. Please see below for detailed replies to all comments. The original review is written in black and our replies in blue.

**A review of the manuscript entitled**

5 **"Drivers of rapid geomagnetic variations at high latitudes",**

**submitted to the journal EGUsphere by Liisa Juusola et al.**

This is a very promising research paper. It exploits the large and comprehensive data resource that is the IMAGE archive. It employs an under-utilized, if not particularly new, analysis technique with 2-layer 2DSECS. It uses these to separate "internal" from "external" equivalent current sources driving geomagnetic disturbance at Earth's surface, and therefore to better under-
10 stand the impact and scale of magnetosphere/ionosphere dynamical phenomena without worrying that "large" geomagnetic disturbances are simply due to much more localized earth conductivity structure. It is also very interesting to see how "internal" sources contribute to the interpretation of space weather phenomena, although perhaps this topic was covered in more detail by a recent paper by the same 1st author, and is only a secondary consideration in the present manuscript. Altogether, this manuscript offers a novel perspective on what influences ground magnetic disturbance the most, for the most impactful
15 space weather events, and therefore should be published and added to the scientific literature base through EGUsphere.

That said, the presentation of this material lacks a certain focus, and is, at times, difficult to read, even for a scientist who is well-acquainted with the analysis techniques and scientific subject matter. The comments below are offered in the constructive hope that the overall clarity and readability of this research paper will be improved, and ultimately appeal to a wider and possibly more scientifically diverse audience.
20 These critiques/recommendations are offered in a loosely prioritized order:

1. There is a considerable review of the underpinning theory in the Introduction. It ends with an overly brief statement of the question being asked/answered, and it is not especially clear how this relates to the material presented prior to that.

   – most of the theory could be migrated into a more fleshed out Section 2.2, including Figures 1 and 2;

   – the introduction could then more clearly and succinctly articulate the motivation behind this study, possibly hinting
25 at the more comprehensive explanation of techniques coming up later.

We suggest to move most of the text at lines 21–60 and Fig. 1–2 to Section 2.2. The sentence "The amplitude of the time derivative of the horizontal ground magnetic field ($|d\boldsymbol{H}/dt|$) has often been used as a proxy for the geoelectric field and GIC risk (Viljanen, 1998; Viljanen et al., 2001)." at lines 21–23 would remain in the Introduction and the expression "external $|d\boldsymbol{H}/dt|$" at line 108 would be replaced by "external (due to ionospheric and magnetospheric currents) $|d\boldsymbol{H}/dt|$".

2. There are too many figures (18!), and many figures include multiple labeled panels, sometimes up to the letter "l" (i.e., 12 panels!). This alone is very distracting, but the real problem is that it is not obvious that all the panels are discussed in the body of the manuscript. The authors should reconsider whether all these are necessary, and if so, could some be migrated to supplementary material. If the authors choose to keep most or all figures, they should make almost all of them larger, probably full-page.

We appreciate the Reviewer's point. It is true that not every single panel is thoroughly discussed for each event. There are two reasons we decided to show all these panels: to make it possible for the reader to compare various features of the different event types, and to avoid the need to introduce new figure layouts for all events. Because of these reasons, we would prefer to keep the panels as they are. However, Fig. 2–3 could be replaced by a reference to the IMAGE webpage, where the same information can be found. We might also remove Fig. 7 and Fig. 8b and the related text section, according to the Reviewer's suggestion below. We would like to keep Fig. 8a, because it demonstrates the important fact that sudden impulses are very fast events, whose details can be partly lost when using 10-s or sparser data. The size of the final figures is probably decided by the copyeditors, but we will try to ensure that they are large enough in case the paper is accepted.

3. There are too many inline mathematical relationships. The authors should consider changing some of these to numbered equations that are visually separated from the main text, then cross-referenced when needed.

We suggest to separate the equations at lines 36 and 121–123 and replace the equation at line 124 by a reference to the equation of line 36.

4. Similarly, there are too many statistics and other data presented inline that would be more clearly presented in numbered, tables, then cross-referenced when needed.

We suggest to replace the statistics at lines 188–191 by a table.

**(some specific comments and questions that should be addressed)**

5. Authors should expand on, or cite specific literature that justifies, the statement in the Introduction: "The down component (Bz) cannot be included in the fitting, because it cannot be represented in terms of ionospheric equivalent currents only".

We suggest to cite Untiedt and Baumjohann (1993) and Vanhamäki and Juusola (2018).

6. Authors should explain, or cite relevant literature justifying, why the internal 2DSECS was defined at only 1m depth? This certainly deviates from much of the previous literature (e.g., Pulkkinen et al., 2003 — EPS), and it seems likely to bias results toward nearby geomagnetic measurements.

Induced currents can flow at any depth below the Earth's surface and an internal equivalent current layer that has been placed deeper than this will not be able to model the currents flowing above it. In reply to the comments by Referee 2, we have suggested to add the following text at line 25 (to be moved to section 2.2, according to the Reviewer's suggestion above): "Both current systems are 3D, but they can be replaced by divergence-free currents on two spherical shells (e.g., Haines and Torta, 1994). These equivalent currents produce the same magnetic field at the Earth's surface as the true 3D currents. The locations of the equivalent current layers are based on physical arguments: the upper layer is at 90 km altitude, practically below all currents in space, and the lower layer is just below the Earth's surface to represent all

induced currents which can flow at any depth. The two-dimensional Spherical Elementary Current Systems (2D SECS) (Vanhamäki and Juusola, 2020) (SECS) method is one option for deriving the divergence-free equivalent currents and separating the variation magnetic field into its external and internal parts. It is based on explicit current distributions from which the magnetic field is calculated according to the Maxwell equations. In real-life applications, availability of the measured magnetic field from a finite set of points in a limited area instead of the whole globe causes some uncertainty, as discussed by Vanhamäki and Juusola (2020). Similar issues naturally concern other methods as well, such as those based on spherical harmonics or Fourier analysis." This topic has also been investigated by Juusola et al. (2016) and $1\,\mathrm{m}$ depth was found to produce good results.

7. The authors should explain better how the results presented in Figures 7 and 8, and related discussion about the data's time resolution, tie into discussion of internal and external sources, and localization of $|dH/dt|$. Frankly, while this is an important point, it seems like a topic for a different paper.

    We suggest to remove lines 237–257, Fig. 7–8, and to modify the conclusions and abstract accordingly.

8. All references to the supplemental animations/movies should make it clear that these are supplemental material. If they could by hyperlinked, even better, at least for the online version of this manuscript.

    We suggest to add "supplementary" before any reference to an animation in the text. The hyperlinking is a very good idea, but will probably be up to the copyeditors.

**(typos, grammatical errors, and ambiguities I noticed)**

9. Line 43 — "in order to be able to produce–d– the highly structured..."

    OK

10. Line 73 — "...by solar wind perturbations, or internally." Clarify "internally".

    We suggest to add "inside the magnetosphere" after "internally".

11. Line 87 — "...rapid $dB/dt$ spikes..." — maybe quote $dB/dt$, assuming it is taken from the cited paper, since the authors consistently use $|dH/dt|$ in this manuscript.

    OK

12. Line 375 — "What is noteworthy in our five –of– events is..."

    OK

**References**

Haines, G. V. and Torta, J. M.: Determination of equivalent current sources from spherical cap harmonic models of geomagnetic field variations, Geophysical Journal International, 118, 3, 499–514, https://doi.org/https://doi.org/10.1111/j.1365-246X.1994.tb03981.x, 1994.

95 Juusola, L., Kauristie, K., Vanhamäki, H., and Aikio, A.: Comparison of auroral ionospheric and field-aligned currents derived from Swarm and ground magnetic field measurements, J. Geophys. Res. Space Physics, 121, 9256–9283, https://doi.org/https://doi.org/10.1002/2016JA022961, 2016.

Untiedt, J. and Baumjohann, W.: Studies of polar current systems using the IMS Scandinavian magnetometer array, Space Science Reviews, 63, 245–390, https://doi.org/https://doi.org/10.1007/BF00750770, 1993.

100 Vanhamäki, H. and Juusola, L.: Review of data analysis techniques for estimating ionospheric currents based on MIRACLE and satellite observations, in: Electric Currents in Geospace and Beyond, pp. 409–425, https://doi.org/https://doi.org/10.1002/9781119324522.ch24, 2018.

Vanhamäki, H. and Juusola, L.: Introduction to Spherical Elementary Current Systems, in: Ionospheric Multi-Spacecraft Analysis Tools, pp. 5–33, ISSI Scientific Report Series 17, https://doi.org/https://doi.org/10.1007/978-3-030-26732-2, 2020.

---

## Author Response (AR1)

**Reply to Referee**

Liisa Juusola[1], Ari Viljanen[1], Andrew P. Dimmock[2], Mirjam Kellinsalmi[1], Audrey Schillings[3], and James M. Weygand[4]

[1]Finnish Meteorological Institute, Helsinki, Finland
[2]Swedish Institute of Space Physics, Uppsala, Sweden
[3]Department of Physics, Umeå University, Umeå, Sweden
[4]Department of Earth, Planetary, and Space Sciences, University of California Los Angeles, Los Angeles, CA, USA

**Dear Professor Engebretson,**

thank you for a positive and constructive review of our study. We were happy to carry out all the suggested corrections. Please see below for detailed replies to all comments. The original review is written in black and our replies in blue. The line numbers refer to the revised version of the manuscript.

5 **Review of Juusola et al., Drivers of rapid geomagnetic variations at high latitudes, submitted to EGUSphere, 2022**

**General Comments**

This is a very well written study of five of the strongest geomagnetic variations observed by the IMAGE magnetometer array. It has a very good introductory review section, followed by tables showing the largest $|\Delta \boldsymbol{H}|$ and $|d\boldsymbol{H}/dt|$ at each of the IMAGE sites, separated into total (observed) and external and internal contributions. This is followed by a detailed analysis

10 of five events that produced some of the most intense external $|d\boldsymbol{H}/dt|$ values. The authors provide plausible interpretations for the magnetospheric/ionospheric phenomena that drove these events, and also provide a careful discussion in section 4 of some of the limitations of this study (even though it is based on a large volume of data) and of continuing challenges to the successful prediction of intense (dangerous) $|d\boldsymbol{H}/dt|$ events. It concludes that the relevant scientific community is still far from a full understanding of the detailed physical pathway(s) leading to either modest or extreme $|d\boldsymbol{H}/dt|$ events, much less

15 to the prediction of the time and place where these events will occur.

The content of this paper is of high quality and is certainly appropriate for publication in EGUsphere. This reviewer has only two substantive comments and four more minor comments.

**Specific Comments**

It is strongly suggested that throughout the paper the magnitude of the perturbations in the horizontal magnetic field
20 that are denoted $|\boldsymbol{H}|$ should be replaced by $|\Delta \boldsymbol{H}|$. The magnitude of the total magnetic field or even its horizontal component (in a given coordinate system) is not what is physically important; it is rather the change in its value (during some appropriate time interval).

Re: Done.

Lines 375–391: The manuscript cites a study by Viljanen et al. (2006) that showed peaks in occurrences of large $|d\boldsymbol{H}/dt|$
25 events 5 minutes after both non-storm and storm-time substorm onsets at Sodankylä (63.92° MLAT) and Nurmijärvi

(56.89° MLAT) during 1997 and 1999 (their Figure 3). However, there were no substorm onsets or sudden intensifi-cations of the western electrojet during the five selected events. The authors may wish to contrast the observations of Viljanen et al. (2006) with those of Engebretson et al. (2021), who showed in their Figure 2 plots of maximum $dB/dt$ events (all $> 6$ nT/s) vs. time delay after substorm onsets for five stations in Arctic Canada during 2015 and 2017, with MLATs ranging from 75.2° to 64.7°. There was no significant peak near 5 minutes after onset at any of these stations (the distributions were relatively flat during the first 30 minutes). The distribution at each station had a gradual and extended falloff that was roughly consistent with those shown in most panels of Figure 3 of Viljanen et al. (2006). The Engebretson et al. (2021) study also showed in panels B and C their Figure 11 that postmidnight dB/dt events that occurred greater than 30 minutes after substorm onsets at the lowest latitude station (KJPK, 64.7° MLAT) occurred during periods of gradual increases in the SML index (weakenings of the WEJ).

Engebretson, M. J., Ahmed, L. Y., Pilipenko, V. A., Steinmetz, E. S., Moldwin, M. B., Connors, M. G., et al. (2021). Superposed epoch analysis of nighttime magnetic perturbation events observed in Arctic Canada. Journal of Geophysical Research: Space Physics, 126, e2021JA029465. https://doi.org/10.1029/2021JA029465

Re: Thank you for reminding us of these results. We have added after line 381: "The second alternative agrees with the results of Viljanen et al. (2006) and Engebretson et al. (2021). Viljanen et al. (2006) examined the occurrence of maximum $|d\boldsymbol{H}/dt|$ after substorm onsets at IMAGE stations during 1997 and 1999. They showed that the largest value of $|d\boldsymbol{H}/dt|$ during substorms occurs most probably at about 5 min after the onset at stations with CGM latitude less than 72°. At this time, the amplitude of the westward electrojet increases rapidly. Engebretson et al. (2021) showed the occurrence of maximum "$dB/dt$" events vs. time delay after substorm onset for five stations in Arctic Canada during 2015 and 2017, with MLATs ranging from 75.2° to 64.7°. There was no significant peak near 5 min after onset at any of these stations, and it was suggested that maximum $dB/dt$ events are not typically associated with substorm onsets but times of the most intense westward electrojet. The key difference between these apparently contradictory results is that, whereas Viljanen et al. (2006) examined the occurrence of maximum $|d\boldsymbol{H}/dt|$ after all identified substorm onsets (with average maximum $|d\boldsymbol{H}/dt|$ typically less than 2.5 nT/s), Engebretson et al. (2021) only considered intense $dB/dt$ events with maximum $dB/dt > 6$ nT/s. Thus, although the intensifying westward electrojet after substorm onset may be a typical source of moderate $|d\boldsymbol{H}/dt|$ values (Viljanen et al., 2006), the rarer events with strong $dB/dt > 6$ nT/s tend to occur during times of the most intense westward electrojet (Engebretson et al., 2021). Engebretson et al. (2021) also showed that postmidnight $dB/dt$ events that occurred greater than 30 min after substorm onsets at the lowest latitude station (KJPK, 64.7° MLAT) occurred during periods of gradual weakenings of the westward electrojet. These could be similar to our events with the undulating westward electrojet."

**Technical Corrections**

In line 185, the phrase "optimal temporal development" does not seem appropriate. "Optimal" approximates to "best", so this part of the sentence is confusing.

Re: We have removed "optimal".

Figure 4 needs to be much larger in the final published paper, and some of the fine print in the figure could be moved to the figure caption. Figures 5, 6, 9, 11, 13, 14, 16, and 17 would also be easier to read if they made use of the full width of the available space on a page.

Re: We have removed Fig. 4 as a response to a comment by Referee 3.

In line 285, "possible" should be changed to "possibly."

Re: Done

In line 375, remove "of" after "five."

Re: Done

**References**

Engebretson, M. J., Ahmed, L. Y., Pilipenko, V. A., Steinmetz, E. S., Moldwin, M. B., Connors, M. G., Boteler, D. H., Weygand, J. M., Coyle, S., Ohtani, S., Gjerloev, J., and Russell, C. T.: Superposed epoch analysis of nighttime magnetic perturbation events observed in Arctic Canada, J. Geophys. Res.: Space Physics, 126, e2021JA029 465, https://doi.org/https://doi.org/10.1029/2021JA029465, 2021.

Viljanen, A., Tanskanen, E. I., and Pulkkinen, A.: Relation between substorm characteristics and rapid temporal variations of the ground magnetic field, Ann. Geophys., 24, 725–733, https://doi.org/https://doi.org/10.5194/angeo-24-725-2006, 2006.

70

**Reply to Referee**

Liisa Juusola[1], Ari Viljanen[1], Andrew P. Dimmock[2], Mirjam Kellinsalmi[1], Audrey Schillings[3], and James M. Weygand[4]

[1]Finnish Meteorological Institute, Helsinki, Finland
[2]Swedish Institute of Space Physics, Uppsala, Sweden
[3]Department of Physics, Umeå University, Umeå, Sweden
[4]Department of Earth, Planetary, and Space Sciences, University of California Los Angeles, Los Angeles, CA, USA

**Dear Reviewer,**

thank you for a critical view-point. We have done our best to reply to each concern. The original review is written in black and our replies in blue. The line numbers refer to the revised version of the manuscript.

5  I am sceptical of the authors' measures of internal and external geomagnetic field variation.

I am happy to be corrected on this, but I see no reason why the variational data could not, in principle, be fitted with a geomagnetic field model that are entirely generated by external current systems. When I say this, of course, I am not referring to the internal field generated in the Earth's core. That part of the field is essentially steady over the course of a magnetic storm. To emphasize, I'm referring to the storm-time variation in the geomagnetic field. I see no reason why that part of the field can't

10  be entirely modelled by external sources.

Re: The horizontal part of the geomagnetic variation field can indeed be expressed in terms of external currents only. However, the vertical ($B_z$) component of the field cannot, as demonstrated by Figure 1 and Figure 2, and indicates that such a reconstruction is not correct.

In this context, recall that a spherical-harmonic description of a global field has both internal and external parts (the division

15  between the two comes with different radial functions), and this dichotomy is consistent with potential-field theory. That sort of internal-external division is rigorous, but such a division is not, to my knowledge, available for spherical elementary currents, where one starts off assuming that internal currents reside on a shell at an arbitrarily chosen depth.

Therefore, I would be extremely careful in making physical interpretations of the *internal* field while relying on an unrealistic assumption about the source of the internal field. The danger of circular reasoning, here, should be clear.

20  I note that the authors seem to admit most this on lines 248–250: "However, separation and interpolation of the geomagnetic field between the stations are not perfect and are affected by the density of the magnetometers as well as boundary conditions, as discussed by Juusola et al. (2020)." One wonders, then, how much the separation can be affected by the boundary conditions. This big issue is not addressed anywhere as far as I know.

My understanding is that the boundary conditions used in spherical elementary current systems are just mathematical conve-

25  niences. As such, they allow construction of specific field models, but these field models are only examples from the large set of models that can fit the data. In other words, the models are non-unique. Some of the possible models (with different chosen boundary conditions) might have lots of internal contribution, but others might have very little.

This sort of non-uniqueness is why spherical elementary currents are often described as "equivalent".

As long as such non-uniqueness exists, I don't know how the authors can come to any conclusions about the relative portions

30  of internal and external fields.

[Figure]

**Figure 1.** Observed magnetic field ("Total") in black and the 2D SECS reconstruction ("External") in blue using a single layer of SECSs at 90 km altitude. The black curve is not visible in panels a, b, d, and f, because it is perfectly covered by the blue curve. The time indicated by the dashed vertical line corresponds to the example provided in the manuscript Figures 2a and 2d.

Re: The total variation magnetic field at the Earth's surface is produced by currents in space (ionosphere and magnetosphere) and in the conducting ground (telluric currents). Both current systems are 3D, but they can be replaced by divergence-free currents on two spherical shells (e.g., Haines and Torta, 1994). These equivalent currents produce the same magnetic field at the Earth's surface as the true 3D currents. The location of the equivalent current layers is based on physical arguments: the upper layer is at 90 km altitude, practically below all currents in space, and the lower layer is just below the Earth's surface to represent all induced currents which can flow at any depth.

The SECS method is one option for deriving the divergence-free equivalent currents and separating the variation magnetic field into its external and internal parts. It is based on explicit current distributions from which the magnetic field is calculated according to the Maxwell equations. So the SECS method is as rigorous and unique as techniques based on spherical harmonics or Fourier analysis, for example, and it has certain practical advantages (Amm and Viljanen, 1999). In real-life applications,

[Figure]

**Figure 2.** Observed magnetic field ("Total") in black and the 2D SECS reconstruction in blue ("External") and red ("Internal") using one layer of SECSs at 90 km altitude and another at 1 m depth. Summing up the blue and red curve produces a perfect match to the black curve in all panels. The time indicated by the dashed vertical line corresponds to the example provided in the manuscript Figures 2b, 2c, 2e, and 2f.

availability of the measured magnetic field from a finite set of points on a limited area instead of the whole globe causes some uncertainty as discussed by Vanhamäki and Juusola (2020). Similar issues naturally concern other methods too. We have shown in a previous study (Juusola et al., 2016) that the ionospheric divergence-free currents produced using the separation correspond very well to currents derived independently from low-orbit satellite-based magnetic field measurements. If the separation is not carried out, the correspondence is clearly worse.

It is true that we should be very careful about making physical interpretations about the 3D currents based on the reconstructed equivalent currents, but the magnetic field as decribed by the equivalent currents between the current sheets should be fairly reliable. An analysis concerning the uncertainties of the reconstruction would be very useful indeed. However, such an analysis is out of the scope of the present study, because it is far from straightforward and would require careful attention, preferably in a dedicated study.

Please consider these issues.

Re: In order to clarify these issues, we have added to line 108:

"Both current systems are 3D, but they can be replaced by divergence-free currents on two spherical shells (e.g., Haines and Torta, 1994). These equivalent currents produce the same magnetic field at the Earth's surface as the true 3D currents. The locations of the equivalent current layers are based on physical arguments: the upper layer is at 90 km altitude, practically below all currents in space, and the lower layer is just below the Earth's surface to represent all induced currents which can flow at any depth. The two-dimensional Spherical Elementary Current System (2D SECS) method (Amm, 1997; Amm and Viljanen, 1999; Pulkkinen et al., 2003a, b; Juusola et al., 2016; Vanhamäki and Juusola, 2020; Juusola et al., 2020) is one option for deriving the divergence-free equivalent currents and separating the variation magnetic field into its external and internal parts. It is based on explicit current distributions from which the magnetic field is calculated according to the Maxwell equations. In real-life applications, availability of the measured magnetic field from a finite set of points in a limited area instead of the whole globe causes some uncertainty, as discussed by Vanhamäki and Juusola (2020). Similar issues naturally concern other methods as well, such as those based on spherical harmonics or Fourier analysis." We have also added references (Untiedt and Baumjohann, 1993; Vanhamäki and Juusola, 2018) for the statement "$B_z$ cannot be included in the fitting, because it cannot be represented in terms of ionospheric equivalent currents only" (lines 133–134).

**References**

Amm, O.: Ionospheric elementary current systems in spherical coordinates and their application, J. Geomagn. Geoelectr., 49, 947–955, https://doi.org/https://doi.org/10.5636/jgg.49.947, 1997.

Amm, O. and Viljanen, A.: Ionospheric disturbance magnetic field continuation from the ground to ionosphere using spherical elementary current systems, Earth Planets Space, 51, 431–440, https://doi.org/https://dx.doi.org/10.1186/BF03352247, 1999.

Haines, G. V. and Torta, J. M.: Determination of equivalent current sources from spherical cap harmonic models of geomagnetic field variations, Geophysical Journal International, 118, 3, 499–514, https://doi.org/https://doi.org/10.1111/j.1365-246X.1994.tb03981.x, 1994.

Juusola, L., Kauristie, K., Vanhamäki, H., and Aikio, A.: Comparison of auroral ionospheric and field-aligned currents derived from Swarm and ground magnetic field measurements, J. Geophys. Res. Space Physics, 121, 9256–9283, https://doi.org/https://doi.org/10.1002/2016JA022961, 2016.

Juusola, L., Vanhamäki, H., Viljanen, A., and Smirnov, M.: Induced currents due to 3D ground conductivity play a major role in the interpretation of geomagnetic variations, Ann. Geophys., 38, 983–998, https://doi.org/https://doi.org/10.5194/angeo-38-983-2020, 2020.

Pulkkinen, A., Amm, O., Viljanen, A., and BEAR Working Group: Ionospheric equivalent current distributions determined with the method of spherical elementary current systems, J. Geophys. Res., 108(A2), https://doi.org/https://doi.org/10.1029/2001JA005085, 2003a.

Pulkkinen, A., Amm, O., Viljanen, A., and BEAR Working Group: Separation of the geomagnetic variation field on the ground into external and internal parts using the spherical elementary current system method, Earth, Planets and Space, 55, 117–129, https://doi.org/https://doi.org/10.1186/BF03351739, 2003b.

Untiedt, J. and Baumjohann, W.: Studies of polar current systems using the IMS Scandinavian magnetometer array, Space Science Reviews, 63, 245–390, https://doi.org/https://doi.org/10.1007/BF00750770, 1993.

Vanhamäki, H. and Juusola, L.: Review of data analysis techniques for estimating ionospheric currents based on MIRACLE and satellite observations, in: Electric Currents in Geospace and Beyond, pp. 409–425, https://doi.org/https://doi.org/10.1002/9781119324522.ch24, 2018.

Vanhamäki, H. and Juusola, L.: Introduction to Spherical Elementary Current Systems, in: Ionospheric Multi-Spacecraft Analysis Tools, pp. 5–33, ISSI Scientific Report Series 17, https://doi.org/https://doi.org/10.1007/978-3-030-26732-2, 2020.

**Reply to Referee**

Liisa Juusola[1], Ari Viljanen[1], Andrew P. Dimmock[2], Mirjam Kellinsalmi[1], Audrey Schillings[3], and James M. Weygand[4]

[1]Finnish Meteorological Institute, Helsinki, Finland
[2]Swedish Institute of Space Physics, Uppsala, Sweden
[3]Department of Physics, Umeå University, Umeå, Sweden
[4]Department of Earth, Planetary, and Space Sciences, University of California Los Angeles, Los Angeles, CA, USA

**Dear Reviewer,**

thank you for a constructive and thorough review. We are happy to carry most of the suggested corrections. Please see below for detailed replies to all comments. The original review is written in black and our replies in blue. Unless otherwise indicated, the line numbers refer to the revised version of the manuscript.

5 **A review of the manuscript entitled**

**"Drivers of rapid geomagnetic variations at high latitudes",**

**submitted to the journal EGUsphere by Liisa Juusola et al.**

This is a very promising research paper. It exploits the large and comprehensive data resource that is the IMAGE archive. It employs an under-utilized, if not particularly new, analysis technique with 2-layer 2DSECS. It uses these to separate "internal"
10 from "external" equivalent current sources driving geomagnetic disturbance at Earth's surface, and therefore to better understand the impact and scale of magnetosphere/ionosphere dynamical phenomena without worrying that "large" geomagnetic disturbances are simply due to much more localized earth conductivity structure. It is also very interesting to see how "internal" sources contribute to the interpretation of space weather phenomena, although perhaps this topic was covered in more detail by a recent paper by the same 1st author, and is only a secondary consideration in the present manuscript. Altogether,
15 this manuscript offers a novel perspective on what influences ground magnetic disturbance the most, for the most impactful space weather events, and therefore should be published and added to the scientific literature base through EGUsphere.

   That said, the presentation of this material lacks a certain focus, and is, at times, difficult to read, even for a scientist who is well-acquainted with the analysis techniques and scientific subject matter. The comments below are offered in the constructive hope that the overall clarity and readability of this research paper will be improved, and ultimately appeal to a wider and
20 possibly more scientifically diverse audience.

   These critiques/recommendations are offered in a loosely prioritized order:

1. There is a considerable review of the underpinning theory in the Introduction. It ends with an overly brief statement of the question being asked/answered, and it is not especially clear how this relates to the material presented prior to that.

   – most of the theory could be migrated into a more fleshed out Section 2.2, including Figures 1 and 2;

25   – the introduction could then more clearly and succinctly articulate the motivation behind this study, possibly hinting at the more comprehensive explanation of techniques coming up later.

Re: We have moved most of the text at lines 21–60 of the previous version of the manuscript and Fig. 1–2 to Section 2.2.

2. There are too many figures (18!), and many figures include multiple labeled panels, sometimes up to the letter "l" (i.e., 12 panels!). This alone is very distracting, but the real problem is that it is not obvious that all the panels are discussed in the body of the manuscript. The authors should reconsider whether all these are necessary, and if so, could some be migrated to supplementary material. If the authors choose to keep most or all figures, they should make almost all of them larger, probably full-page.

Re: We appreciate the Reviewer's point. It is true that not every single panel is thoroughly discussed for each event. There are two reasons we decided to show all these panels: to make it possible for the reader to compare various features of the different event types, and to avoid the need to introduce new figure layouts for all events. Because of these reasons, we would prefer to keep the panels as they are. However, we have replaced Fig. 3–4 of the previous version of the manuscript by a reference to the IMAGE webpage, where the same information can be found. We have also remove Fig. 7 and Fig. 8b and the related text section, according to the Reviewer's suggestion below. We would like to keep Fig. 8a (Fig. 14 in the new version), because it demonstrates the important fact that sudden impulses are very fast events, whose details can be partly lost when using 10-s or sparser data. We have increased the size of the remaining figures, but the final appearance is probably decided by the copyeditors.

3. There are too many inline mathematical relationships. The authors should consider changing some of these to numbered equations that are visually separated from the main text, then cross-referenced when needed.

Re: We have separated the equations in section 2.1 and used cross-references in the following text.

4. Similarly, there are too many statistics and other data presented inline that would be more clearly presented in numbered, tables, then cross-referenced when needed.

Re: We have replaced the statistics at lines 188–191 of the previous version of the manucript by a table (Table 2).

**(some specific comments and questions that should be addressed)**

5. Authors should expand on, or cite specific literature that justifies, the statement in the Introduction: "The down component (Bz) cannot be included in the fitting, because it cannot be represented in terms of ionospheric equivalent currents only".

Re: We have added references to Untiedt and Baumjohann (1993) and Vanhamäki and Juusola (2018).

6. Authors should explain, or cite relevant literature justifying, why the internal 2DSECS was defined at only 1m depth? This certainly deviates from much of the previous literature (e.g., Pulkkinen et al., 2003 — EPS), and it seems likely to bias results toward nearby geomagnetic measurements.

Re: Induced currents can flow at any depth below the Earth's surface and an internal equivalent current layer that has been placed deeper than this will not be able to model the currents flowing above it. In reply to the comments by Referee 2, we have added the followed text at line 108: "Both current systems are 3D, but they can be replaced by divergence-free currents on two spherical shells (e.g., Haines and Torta, 1994). These equivalent currents produce the same magnetic field at the Earth's surface as the true 3D currents. The locations of the equivalent current layers are based on physical arguments: the upper layer is at 90 km altitude, practically below all currents in space, and the lower layer is just below the Earth's surface to represent all induced currents which can flow at any depth. The two-dimensional Spherical Elementary Current System (2D SECS) method (Amm, 1997; Amm and Viljanen, 1999; Pulkkinen et al., 2003a, b; Juusola et al., 2016; Vanhamäki and Juusola, 2020; Juusola et al., 2020) is one option for deriving the divergence-free equivalent currents and separating the variation magnetic field into its external and internal parts. It is based on explicit current distributions from which the magnetic field is calculated according to the Maxwell equations. In real-life

applications, availability of the measured magnetic field from a finite set of points in a limited area instead of the whole globe causes some uncertainty, as discussed by Vanhamäki and Juusola (2020). Similar issues naturally concern other methods as well, such as those based on spherical harmonics or Fourier analysis." This topic has also been investigated by Juusola et al. (2016) and 1 m depth was found to produce good results. We have added this reference to line 121.

7. The authors should explain better how the results presented in Figures 7 and 8, and related discussion about the data's time resolution, tie into discussion of internal and external sources, and localization of $|dH/dt|$. Frankly, while this is an important point, it seems like a topic for a different paper.

Re: We have removed Fig. 7 and 8b and the related text sections. Fig. 8a (currently Fig. 14), was moved to the Discussion (section 4.1).

8. All references to the supplemental animations/movies should make it clear that these are supplemental material. If they could by hyperlinked, even better, at least for the online version of this manuscript.

Re: We have added "supplementary" before any reference to an animation in the text. The hyperlinking is a very good idea, but will probably be up to the copyeditors.

**(typos, grammatical errors, and ambiguities I noticed)**

9. Line 43 — "in order to be able to produce–d– the highly structured..."

Done.

10. Line 73 — "...by solar wind perturbations, or internally." Clarify "internally".

We have added "inside the magnetosphere" after "internally".

11. Line 87 — "...rapid $dB/dt$ spikes..." — maybe quote $dB/dt$, assuming it is taken from the cited paper, since the authors consistently use $|dH/dt|$ in this manuscript.

Done.

12. Line 375 — "What is noteworthy in our five –of– events is..."

Done.

**References**

Amm, O.: Ionospheric elementary current systems in spherical coordinates and their application, J. Geomagn. Geoelectr., 49, 947–955, https://doi.org/https://doi.org/10.5636/jgg.49.947, 1997.

Amm, O. and Viljanen, A.: Ionospheric disturbance magnetic field continuation from the ground to ionosphere using spherical elementary current systems, Earth Planets Space, 51, 431–440, https://doi.org/https://dx.doi.org/10.1186/BF03352247, 1999.

Haines, G. V. and Torta, J. M.: Determination of equivalent current sources from spherical cap harmonic models of geomagnetic field variations, Geophysical Journal International, 118, 3, 499–514, https://doi.org/https://doi.org/10.1111/j.1365-246X.1994.tb03981.x, 1994.

Juusola, L., Kauristie, K., Vanhamäki, H., and Aikio, A.: Comparison of auroral ionospheric and field-aligned currents derived from Swarm and ground magnetic field measurements, J. Geophys. Res. Space Physics, 121, 9256–9283, https://doi.org/https://doi.org/10.1002/2016JA022961, 2016.

Juusola, L., Vanhamäki, H., Viljanen, A., and Smirnov, M.: Induced currents due to 3D ground conductivity play a major role in the interpretation of geomagnetic variations, Ann. Geophys., 38, 983–998, https://doi.org/https://doi.org/10.5194/angeo-38-983-2020, 2020.

Pulkkinen, A., Amm, O., Viljanen, A., and BEAR Working Group: Ionospheric equivalent current distributions determined with the method of spherical elementary current systems, J. Geophys. Res., 108(A2), https://doi.org/https://doi.org/10.1029/2001JA005085, 2003a.

Pulkkinen, A., Amm, O., Viljanen, A., and BEAR Working Group: Separation of the geomagnetic variation field on the ground into external and internal parts using the spherical elementary current system method, Earth, Planets and Space, 55, 117–129, https://doi.org/https://doi.org/10.1186/BF03351739, 2003b.

Untiedt, J. and Baumjohann, W.: Studies of polar current systems using the IMS Scandinavian magnetometer array, Space Science Reviews, 63, 245–390, https://doi.org/https://doi.org/10.1007/BF00750770, 1993.

Vanhamäki, H. and Juusola, L.: Review of data analysis techniques for estimating ionospheric currents based on MIRACLE and satellite observations, in: Electric Currents in Geospace and Beyond, pp. 409–425, https://doi.org/https://doi.org/10.1002/9781119324522.ch24, 2018.

Vanhamäki, H. and Juusola, L.: Introduction to Spherical Elementary Current Systems, in: Ionospheric Multi-Spacecraft Analysis Tools, pp. 5–33, ISSI Scientific Report Series 17, https://doi.org/https://doi.org/10.1007/978-3-030-26732-2, 2020.

---

## Referee Report (RR1)

Review of Juusola et al., Drivers of rapid geomagnetic variations at high latitudes, submitted to EGUSphere, 2022, revised version.

The authors have successfully addressed all of this reviewer's comments on the originally submitted manuscript, and other changes in the paper have improved its organization and readability. The paper should be an important contribution to the literature. I recommend acceptance.